# Evolution of a truncated nucleocapsid protein enhances SARS-CoV-2 fitness by suppressing antiviral responses

Rory P. Mulloy[1], Danyel Evseev[1], Noga Sharlin[1], Maxwell P. Bui-Marinos[1], Émile Lacasse[2], Isabelle Dubuc[2], Louis Flamand[2,3], Jennifer A. Corcoran [ID][1‡*]

1 Microbiology, Immunology and Infectious Diseases Department, Snyder Institute for Chronic Diseases, Charbonneau Cancer Research Institute, Cumming School of Medicine, University of Calgary, Calgary, Alberta, Canada, 2 Axe Maladies Infectieuses et Immunitaires, Centre de Recherche du CHU de Québec-Université Laval, Québec, Canada, 3 Department of Microbiology, Infectious Disease and Immunology, Faculty of Medicine, Université Laval, Québec, Canada

‡ Lead contact.
* jennifer.corcoran@ucalgary.ca

## Abstract

Viruses face selective pressure to evade cellular antiviral responses to control the outcome of an infection. However, due to their limited genome size, viruses must adopt unique strategies to confront cellular sensors. Since its emergence in humans, SARS-CoV-2 accrued many mutations; however, the functional consequence of many such genetic changes remains unexplored. Here, we show that SARS-CoV-2 produces a truncated form of the nucleocapsid protein, called $N^{*M210}$. Due to the acquisition of a viral transcription regulatory sequence (TRS) in the N gene, certain variants like Omicron produce a new viral mRNA that markedly increases $N^{*M210}$ production. We show that $N^{*M210}$ is a double-stranded RNA (dsRNA)-binding protein. Using its dsRNA binding motif, $N^{*M210}$ inhibits multiple antiviral responses, supressing interferon, triggering processing body disassembly, and potently blocking G3BP1 foci, including stress granules and RNase L-dependent bodies. Using a panel of recombinant SARS-CoV-2 viruses (rSARS-2), we show that enhanced $N^{*M210}$ production increases virus fitness in primary human cells and in mice. Furthermore, we show that during infection $N^{*M210}$ improves virus fitness, in part, due to its ability to potently block G3BP1 foci. We propose a model where, to evade the cellular antiviral response, SARS-CoV-2 has evolved a mechanism to increase the production of a truncated form of the N protein, which limits activation of dsRNA-induced antiviral responses, tipping the balance in favor of the virus in the battle for control of the cell.

## Introduction

The outcome of a viral infection is determined by two factors: the ability of the cell to sense and respond to the virus, and the ability of the virus to evade these responses.

which permits unrestricted use, distribution, and reproduction in any medium, provided the original author and source are credited.

**Data availability statement:** Recombinant SARS-CoV-2 sequences can be accessed on NCBI: accession numbers: WT: PV935594.1, KR+TRS: PV952734, KR−TRS: PV936453, M210I: PV955084. Rscript pipeline for SARS-CoV-2 mutational analysis (Figs 9 and S6) can be accessed on github https://github.com/rmulloy97/Mulloy-et-al.-SARS-CoV-2-TRS-prevalence or Zenodo.org DOI 10.5281/zenodo.18225476. All other data underlying the findings in this study are available in Supporting information (S1 Supporting Data and S1 Data).

**Funding:** RPM was supported by a Snyder Institute Beverley Phillips Doctoral training award and a Doctoral Canada Graduate scholarship from the Canadian Institutes of Health Research. MPBM was supported by a Cumming School of Medicine graduate training award, a Snyder Institute Beverley Phillips Doctoral training award and an Alberta graduate scholarship. NS was supported by a Master's Canada Graduate scholarship from the Canadian Institutes of Health Research and a Cumming School of Medicine graduate training award. This study was supported by operating funds awarded to JAC from the Institute of Infection and Immunity of the Canadian Institutes for Health Research (https://cihr-irsc.gc.ca/e/13533.html; project grant #195645) and received partial support from an operating grant (#175622) awarded to the Coronavirus Variants Rapid Response Network (CoVaRR-Net; https://covarrnet.ca/) from the Institute of Infection and Immunity of the Canadian Institutes of Health Research (https://cihr-irsc.gc.ca/e/13533.html). JAC and LF were members of CoVaRR-Net. The funders had no role in study design, data collection and analysis, decision to publish, or preparation of the manuscript.

**Competing interests:** The authors have declared that no competing interests exist.

**Abbreviations:** BAC, bacterial artificial chromosome; B-TRS, body TRS; COVID-19, coronavirus disease 2019; CoVs, coronaviruses; CTD, C-terminal domain; dsRBMdsRNA binding motif; dsRNA, double-stranded RNA; EV, empty vector; G3BP1, GTPase-activating RNA-binding protein 1; IFN, interferons; IRF3,

Through this evolutionary conflict, cells have developed numerous mechanisms to restrict viral replication, and in turn viruses have acquired methods to suppress these antiviral responses to promote their replication and spread. Many viruses, including coronaviruses (CoVs), produce double-stranded RNA (dsRNA) during infection, activating multiple antiviral responses [1,2]. dsRNA sensing induces translational arrest and the production of type I interferons (IFN), triggering a paracrine and autocrine induction of an antiviral state [3]. More recently, antiviral programmes that rely on the formation of ribonucleoprotein (RNP) granules have been described, which include processing bodies (P-bodies) and G3BP1 foci, such as stress granules (SGs) and RNase L-dependent bodies (RLBs) [4–7]. dsRNA can induce both interferon production and the formation of RLBs. Although this is a rapidly emerging field, the precise mechanism for how these RNP granules antagonize virus replication remains incompletely understood. What is clear is that a diversity of viruses, including CoVs, inhibit G3BP1 foci (SGs and RLBs) and P-bodies during infection [8–17]. Both interferon and antiviral granules impose a barrier to virus replication, explaining why viruses have evolved mechanisms to evade or dampen these antiviral programmes.

Severe-acute respiratory syndrome CoV-2 (SARS-CoV-2) is an RNA virus and the causative agent of coronavirus disease 2019 (COVID-19). Despite concerted efforts to produce vaccines and antivirals, due to its evolutionary dexterity, SARS-CoV-2 still poses a significant health risk. Following its emergence and circulation in the human population, SARS-CoV-2 accrued a multitude of mutations throughout its genome, some of which have improved virus fitness, enabling new genotypes to outcompete the ancestral virusand driving the rise of viral variants [18,19]. While much recent attention has been focused on changes in the spike gene, many other SARS-CoV-2 proteins contribute to viral transmission, pathology, and overall virus fitness; however, the biological consequences of most of these changes remain undiscovered. One viral gene subject to significant evolutionary change is the nucleocapsid (N) gene [20–23]. Here, we focus on one genetic change in the N gene and determine how its acquisition impacts SARS-CoV-2 replication, fitness, and antiviral antagonism (Fig 1).

Upon infection, SARS-CoV-2 delivers its 30 kb genome to the cytoplasm where it is directly translated, promoting the synthesis of non-structural proteins (nsps), including the viral RNA polymerase and proteins that form replication organelles [3]. In replication organelles, SARS-CoV-2 transcribes viral messenger RNAs called subgenomic RNAs (sgRNAs) [3]. sgRNA synthesis is governed by the presence of short sequences called transcription regulatory sequences (TRSs) in the genomic RNA (black boxes in Fig 2A) [24–26]. TRSs contain a conserved ~6-nucleotide core sequence. The leader TRS (L-TRS) is located at the 5′ end of the genome, whereas the body TRS (B-TRS) sequences are positioned near the 5′ end of some accessory genes and each structural gene, including N (Fig 2A). Certain SARS-CoV-2 variants, including Alpha, Gamma, and Omicron, have acquired a TRS in the gene body of N, resulting in the synthesis of a new sgRNA [18].

The N protein is a multifunctional protein with essential roles in CoV replication, such as transcription, genome replication, and packaging [27,28]. Additionally, N blocks multiple branches of the cellular antiviral response, preventing translation

interferon regulatory factor-3; hpi, hours post-infection; HRI, heme-regulated inhibitor; L-TRS, leader TRS; M, methionine; MGB, minor groove binding; N, nucleocapsid; NFQnon-fluorescent quencher; nsps, non-structural proteins; NTD, N-terminal domain; OAS, 2′-5′-oligoadenylate synthetase; PAF, polymerase II-associated factor; PHAC, Public Health Agency of Canada; PKR, protein kinase R; RLBs, RNase L-dependent bodies; RLRs, RIG-I-like receptors; RNP, ribonucleoprotein; rSARS-2, recombinant SARS-CoV-2 viruses; SA, sodium arsenite; SARS-CoV-2, severe-acute respiratory syndrome CoV-2; sgRNAs, subgenomic RNAs; SGs, stress granules; SR, serine-arginine; TRS, transcription regulatory sequence; WT, wild-type.

arrest, inhibiting interferon production, blocking SGs, and disassembling P-bodies [11,14,29–31]. Structurally, N is a modular protein containing two domains: the N-terminal domain (NTD) and the C-terminal domain (CTD). These domains are flanked and linked by flexible disordered regions (N-IDR, linker, and C-IDR) (Fig 2B). Within the linker, there is a serine-arginine (SR)-rich region that becomes hyperphos-phorylated upon infection [32]. Both the NTD and the CTD contain RNA-binding activity, each with unique specificities; the NTD binds TRS-like sequences with sequence specificity, whereas the CTD has a high affinity for structured and dsRNA without known sequence specificity [33,34].

Here, we show that all variants of SARS-CoV-2 produce multiple truncated versions of N (N*). One truncated N proteoform is made by internal translation initiation at the methionine codon at position 210 of the full-length N sequence and is referred to as N*M210. Moreover, multiple viral variants, including currently circulating Omicron vari-ants, up-regulate N*M210 production due to the acquisition of a canonical TRS within the N gene. This internal TRS permits the transcription of a novel sgRNA, from which N*M210 is efficiently translated. Using a panel of recombinant SARS-CoV-2 viruses (rSARS-2), we reveal that enhanced N*M210 production increases virus fitness in pri-mary human cells and in mice. During the preparation of this manuscript, Mears and colleagues (2025) also showed that the new N sgRNA encodes an amino-terminally truncated form of the N protein and that this mutation enhances viral fitness in a lung cancer cell line [35]. Our work confirms and enhances this study and provides further mechanistic analysis. We reveal that N*M210 sequesters dsRNA and prevents activa-tion of dsRNA-induced antiviral responses. Using co-infection competition assays, we demonstrate that the fitness advantage provided by N*M210 production is, in part, due to its potent ability to block G3BP1 foci formation. By characterizing the function of this viral gene product, we explain how a genetic adaptation likely contributed to the dominance of certain variants in the human population (Fig 1).

## Results

### SARS-CoV-2 produces truncated N proteoforms

The N gene has been under considerable evolutionary pressure, with unique muta-tions arising in all the dominant variants, including Alpha (B.1.1.7), Beta (B.1.351), Gamma (P.1), Delta (B.1.617.2), and Omicron (BA-1). The linker region in N contains a mutational hotspot, resulting in various amino acid changes compared to the ances-tral virus, including R203M in the Delta lineage and R203K/G204R in Alpha, Gamma, and Omicron lineages (Fig 2B and 2C). We and others noticed that the mutation forming the R203K/G204R substitution forms a canonical TRS within the gene body of N (Fig 2C) [18,35]. To test if this internal TRS acquisition enables transcription of a novel sgRNA, we isolated RNA from SARS-CoV-2-infected cells and conducted RT-PCR diagnostics. Using a forward primer complementary to the L-TRS (found at the 5′ end of all sgRNAs) and a reverse primer complementary to sequences in the body of the N gene, PCR amplification should produce a ~800 bp amplicon when hybridized to canonical full-length N sgRNA and a ~200 bp amplicon if hybridized to

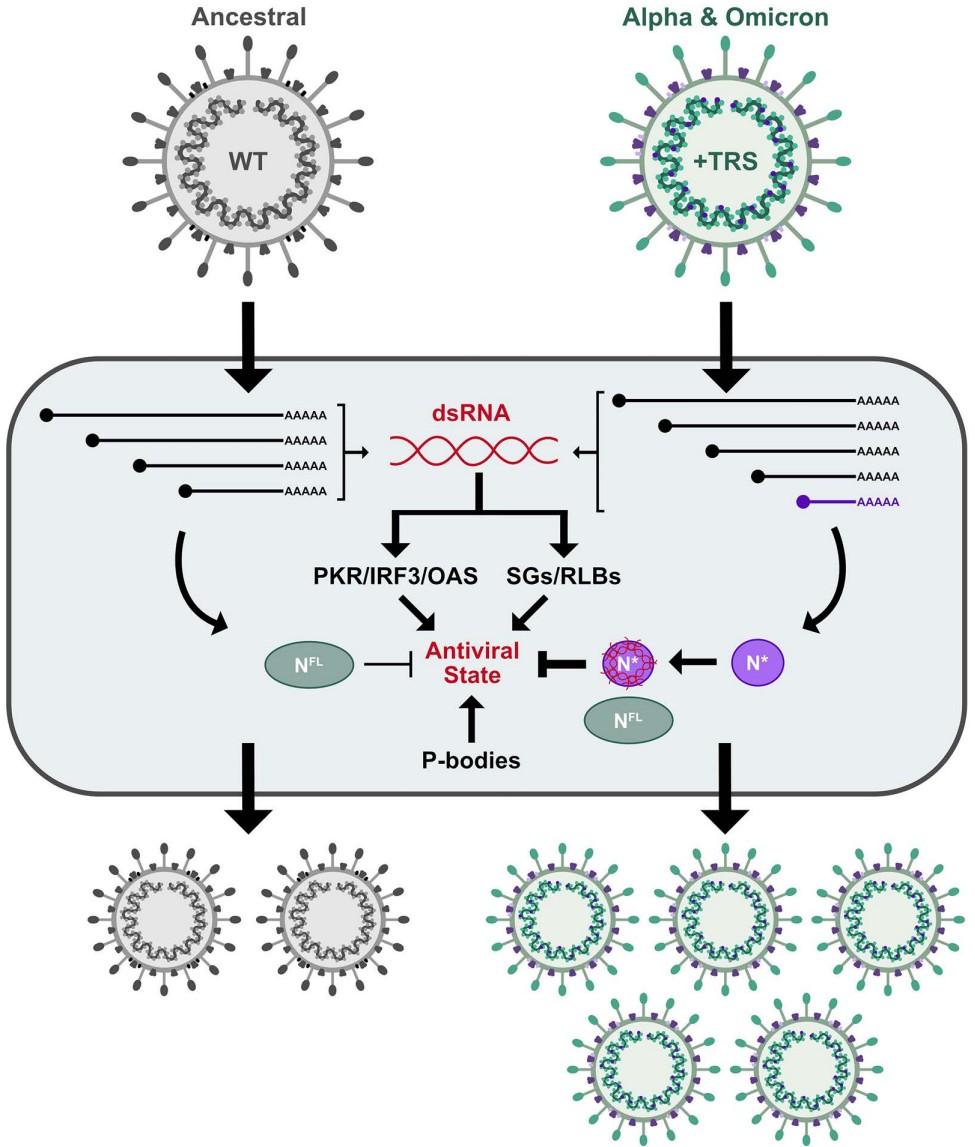

**Fig 1. Evolution of a truncated nucleocapsid protein enhances SARS-CoV-2 fitness by suppressing antiviral responses.** SARS-CoV-2 evolved to acquire a transcription regulatory sequence (TRS) internal to the N gene to direct the transcription of a novel subgenomic RNA, sgRNA-N*, and enhance production of a truncated form of the N protein called N*M210. N*M210 binds dsRNA, blocking multiple antiviral responses including the formation of antiviral ribonucleoprotein granules like stress granules (SGs) and RNaseL-dependent bodies (RLBs). Schematic made using Affinity Designer.

the novel truncated sgRNA (Fig 2A inset). We detected amplicons corresponding to the canonical full-length sgRNA-N from cells infected with all SARS-CoV-2 variants including a Wuhan-like ancestral isolate called TO-1 (Fig 2D and 2E) [36]. However, only variants that contain the internal TRS (Alpha, Gamma, and Omicron) produced the smaller amplicon. These data confirm that the novel TRS promotes the transcription of a new sgRNA, henceforth called sgRNA-N*.

To determine if sgRNA-N* is protein-coding, we looked for open reading frames. The N gene contains several in-frame methionine-coding codons (AUGs) with the potential to produce truncated N proteoforms by internal translation initiation. These methionine (M) codons can be found at positions 101, 210, and 234. M210 and M234 are located immediately

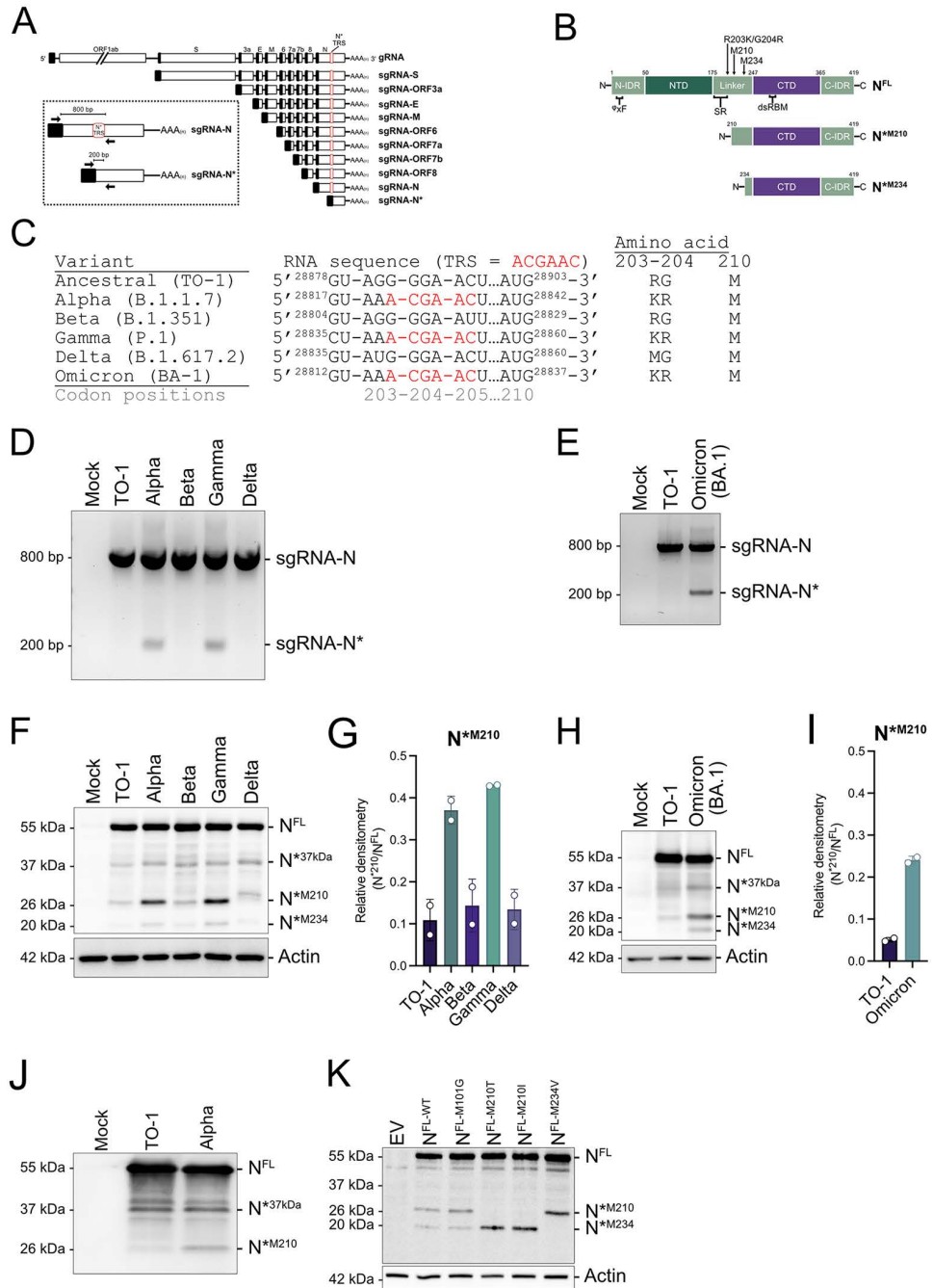

**Fig 2. Select SARS-CoV-2 variants produce truncated proteoforms of the nucleocapsid protein by internal TRS acquisition. A.** A schematic representation of the genomic and subgenomic RNAs with canonical TRSs (black boxes) and the novel TRS (red box). The inset depicts the strategy to detect sgRNA-N* by RT-PCR. The forward primer binds to the 5' end of the leader sequence (40−75 nt) and the reverse primer binds down-stream of the novel TRS (29,035−29,053 nt). **B.** Domain organization of $N^{FL}$ and truncated N proteoforms $N^{*M210}$ and $N^{*M234}$. Amino acid positions are labeled above, relative to nomenclature for $N^{FL}$. Intrinsically disordered region (IDR), N-terminal domain (NTD), and C-terminal domain (CTD), serine-arginine rich area (SR), double-stranded RNA binding motif (dsRBM). φxF (φ is a hydrophobic amino acid; I15–F17) is the G3BP1-binding motif (60). **C.** SARS-CoV-2 variant sequence diversity at the mutational hotspot in the N gene where canonical TRS insertion occurred. Sequence of the core TRS hexamer is labeled in red. Codon positions are based on the ancestral sequence. **D, E.** sgRNA profile of SARS-CoV-2 variants was determined by infecting Calu3 cells (MOI = 2 for D, MOI = 1 for E). RNA was harvested at 24 hpi and subjected to RT-PCR and agarose gel electrophoresis. Toronto-1 (TO-1) isolate is the Wuhan-like ancestral variant. See S1 Supporting Data for full gels. **F–I.** N proteoform profile of SARS-CoV-2 variants was determined by infecting Calu3 cells (MOI = 2 for F and G, MOI = 1 for H and I). Protein lysate was harvested 24 hpi and subjected to SDS-PAGE and immunoblotting with anti-N

and anti-actin antibodies. Protein quantification was conducted by densitometry. N*$^{M210}$ proteoform abundance is presented relative to N$^{FL}$ abundance (N*$^{M210}$/N$^{FL}$). These data represent two independent biological replicates ($n = 2$) and plotted with standard deviation (SD). See S1 Supporting Data for full blots and S1 Data for densitometry values. **J.** N proteoform profile of extracellular SARS-CoV-2 virions. VeroE6 cells were infected with TO-1 or Alpha (MOI = 1). Supernatant was harvested 24 hpi and purified via ultracentrifugation. Concentrated virions were lysed and resolved by SDS-PAGE and immunoblotting. See S1 Supporting Data for full blots. **K.** HEK293Ts were transfected with plasmids encoding a carboxy-terminally FLAG-tagged versions of N$^{FL}$ either without substitution of internal methionine codons (N$^{FL-WT}$) or with the indicated internal methionine codon substitutions or with an empty vector (EV) control. Protein lysates were harvested 24 hours post-transfection and resolved by SDS-PAGE and immunoblotting for FLAG and actin. See S1 Supporting Data for full blots.

downstream of the internal TRS, which denotes the transcription start site for sgRNA-N*, suggesting that they could be more efficiently translated from sgRNA-N* (Fig 2B). To determine if SARS-CoV-2 infection produces truncated N proteoforms, we infected cells with the panel of viral variants and conducted immunoblotting for N. We found that all viruses tested produced multiple lower molecular weight proteoforms of N, including 37, 26, and 20 kDa products (Figs 2F–2I and S1A–S1D). Two pieces of data indicate that the 26 kDa band corresponds to an amino-terminally truncated N proteoform initiating at M210, which will hereafter be referred to as N*$^{M210}$: first, overexpression of a truncated N proteoform starting at M210 yields a 26 kDa product (S1E Fig); second, this product was upregulated by variants that produce sgRNA-N* (Fig 2F and 2H). Variants that do not produce sgRNA-N* (TO-1, Beta, and Delta), still produce N*$^{M210}$ protein, albeit much less abundantly (Fig 2F–2I). This is likely a result of internal ribosomal initiation at the downstream AUG at position M210. Given that N$^{FL}$ is a structural protein, we wondered if N*$^{M210}$ could also be packaged into viral particles. To test this, extracellular virions from TO-1 and Alpha-infected cells were concentrated, lysed, and subjected to immunoblotting. We determined that multiple N* proteoforms, including N*$^{M210}$, can be found in viral particles (Fig 2J).

We also observed a 20 kDa proteoform was produced by all variants tested but upregulated in cells infected with internal TRS-containing viruses (Figs 2F, 2H, S1A, and S1B). This product likely derives from internal translation initiation at the AUG codon M234 (S1E and S1F Fig); hereafter, this N proteoform will be referred to as N*$^{M234}$. N*$^{M234}$ is minimally produced by infection with viruses that do not produce the sgRNA-N*, likely via internal ribosomal initiation from the sgRNA-N transcript. However, viruses that produce sgRNA-N* (Alpha, Gamma, Omicron) have increased N*$^{M234}$ expression, likely via internal translation initiation from the shorter sgRNA-N* transcript. When N$^{FL}$ was ectopically expressed in HEK293T cells, we could detect a similar array of lower molecular weight proteoforms, including N*$^{M210}$ and N*$^{M234}$, supporting the hypothesis that N* products can be produced by internal translation initiation (Fig 2K). The 37 kDa product (termed N*$^{37kDa}$) was not detected during ectopic expression. Substitution of M210 and M234 in the N$^{FL}$ sequence caused the disappearance of N*$^{M210}$ and N*$^{M234}$, respectively (Fig 2K). Substitution of M101 had no effect on the production of any truncated N proteoforms. These data suggest that N*$^{M210}$ and N*$^{M234}$, but not N*$^{37kDa}$, are produced by internal ribosomal initiation. To uncouple the production of N$^{FL}$ from N* during ectopic expression experiments, we mutated downstream initiation codons in N$^{FL}$ (N$^{FL-M210I/M234V}$) and N*$^{M210}$ (N*$^{M210-M234V}$) for all subsequent experiments unless otherwise noted (S1E and S1F Fig). To determine the kinetics of sgRNA-N* transcription and N* protein production, Calu3 cells were infected with the Alpha variant. We observed that sgRNA-N and sgRNA-N* were produced concurrently, with transcription initiating between 2 and 4 hours post-infection (hpi) and N$^{FL}$ and N*$^{M210}$ proteins first detected between 4 and 8 hpi (S1G Fig). Taken together, these data show that SARS-CoV-2 produces truncated proteoforms (N*) in two distinct ways: (i) by internal translation initiation from in-frame AUGs; and (ii) by the synthesis of a novel sgRNA in select variants. Given that some SARS-CoV-2 variants acquired the ability to produce increased amounts of truncated N proteoforms, especially N*$^{M210}$, we asked, does N* production increase virus fitness?

### Increased N*$^{M210}$ production confers a viral fitness advantage in vitro and in vivo

Prior studies reported that the R203K/G204R substitution enhanced viral fitness by increasing N$^{FL}$ phosphorylation [20,23]. However, this mutation also promotes the synthesis of sgRNA-N*, which increases the production of N*$^{M210}$ (Fig 2). Using

   

the ancestral wild-type (WT) backbone [37,38], we constructed a panel of rSARS-2 designed to uncouple the effects of the R203K/G204R amino acid change from the insertion of the novel TRS. First, we cloned a virus containing the R203K/G204R substitution with the authentic nucleotide sequence, resulting in the insertion of the canonical TRS; this virus was named KR$^{+TRS}$ (Fig 3A). Next, we created a virus containing the R203K/G204R substitution without the introduction of a TRS utilizing the degeneracy of the amino acid code; this virus was named KR$^{-TRS}$. To block all N*$^{M210}$ production via internal ribosome initiation, we also substituted the N*$^{M210}$ start codon in the WT backbone; this virus was named M210I. Each recombinant virus was validated by whole-genome sequencing. With this rSARS2 panel, we aimed to test if N*$^{M210}$ synthesis could influence viral fitness.

We used diagnostic RT-PCR to validate that insertion of the TRS in the KR$^{+TRS}$ virus promoted sgRNA-N* synthesis. Only cells infected with KR$^{+TRS}$ produced sgRNA-N*, whereas all recombinants produced the canonical full-length sgRNA-N (Figs 3B and S2A). We also observed that cells infected with KR$^{+TRS}$ produced N*$^{M210}$ and N*$^{M234}$ in greater abundance than cells infected with WT and KR$^{-TRS}$ recombinants, while M210I did not produce any N*$^{M210}$ (Figs 3C, 3D, and S2B–S2G). The inability to detect the N*$^{M210}$ proteoform from M210I-infected cells strongly suggests that N*$^{M210}$ is produced via internal ribosomal initiation and not by proteolytic cleavage, as previously suggested [39]. We observed that the 37 kDa proteoform (N*$^{37kDa}$) was produced in cells infected with all recombinants, though it was more abundant in cells infected with KR$^{+TRS}$ and KR$^{-TRS}$ (Figs 3C, S2C, S2D, and S2G). N*$^{37kDa}$ may represent a proteolytic cleavage product of N$^{FL}$ (see Discussion).

To test if increased production of N*$^{M210}$ provides a fitness advantage, we performed competition assays between recombinant viruses that make more N*$^{M210}$ (KR$^{+TRS}$) compared to viruses that make less (WT, KR$^{-TRS}$, and M210I). Recombinant virus species were added in a 1:1 ratio based on infectious titer to A549$^{ACE2}$ cells and primary endothelial cells (HUVEC$^{ACE2}$) (Fig 3E). Intracellular RNA was harvested at various times post-infection. The abundance of each recombinant virus species was quantified using a probe-based RT-qPCR assay where the probe binding site spans the mutated region of the TRS or M210I substitution, thereby allowing each recombinant virus to be differentiated in a mixed population and the relative proportion of each viral RNA species to be determined (S2H Fig). Probes were validated for specificity with intracellular RNA extracted from cells independently infected with each recombinant virus (S2I Fig).

In both A549$^{ACE2}$ and HUVEC$^{ACE2}$ cells, KR$^{+TRS}$ readily outcompeted the WT virus (Fig 3F and 3I). To discern whether the fitness advantage derived from the amino acid change R203K/G204R or the internal TRS nucleotide acquisition, which are both present in the KR$^{+TRS}$ virus, we next conducted competition assays between KR$^{+TRS}$ and KR$^{-TRS}$ recombinant viruses. These viruses produce identical N$^{FL}$ proteins but differ in the abundance of N*$^{M210}$ that is made during infection (Figs 3C and S2D). In both A549$^{ACE2}$ and HUVEC$^{ACE2}$ cells, the KR$^{+TRS}$ virus outcompeted the KR$^{-TRS}$ virus, suggesting that greater N*$^{M210}$ production confers a fitness advantage (Fig 3G and 3J). Given that N*$^{M210}$ can be basally produced by internal ribosome initiation at M210 by both WT and KR$^{-TRS}$ viruses, we wondered if even minimal N*$^{M210}$ expression could influence virus fitness. To test this, we performed competition assays between WT and M210I viruses. However, we detected no significant fitness differences between these viruses, suggesting that basal production of N*$^{M210}$ does not significantly alter viral fitness in vitro (Fig 3H and 3K). Analysis of extracellular RNA, which includes viral genomes from mature viral particles, revealed similar trends where KR$^{+TRS}$ virus outcompeted both WT and KR$^{-TRS}$ viruses, and WT and M210I were not statistically different (S2J–S2L Fig). We found no significant fitness difference between WT and KR$^{-TRS}$ viruses in HEK293A$^{ACE2}$ cells (Fig 3L). Both WT and KR$^{-TRS}$ lack the N* TRS and produce low levels of N*$^{M210}$, but these viruses produce different N$^{FL}$ proteins; unlike WT, the KR$^{-TRS}$ virus produces N$^{FL}$ containing the amino acid change R203K/G204R. This suggests that the N* TRS acquisition, rather than the N$^{FL}$ amino acid change, is the primary driver of the fitness difference. We also show that the TRS mutation confers a fitness advantage in vivo, as the KR$^{+TRS}$ virus readily outcompeted KR$^{-TRS}$ virus upon co-infection of K18$^{ACE2}$ mice (Figs 3M and S2M). Taken together, these data indicate that viruses that produce N*$^{M210}$ in greater abundance have enhanced fitness in lung epithelial cancer cells, in primary endothelial cells, and in vivo, yet the mechanism by which N*$^{M210}$ promotes virus replication is unknown.

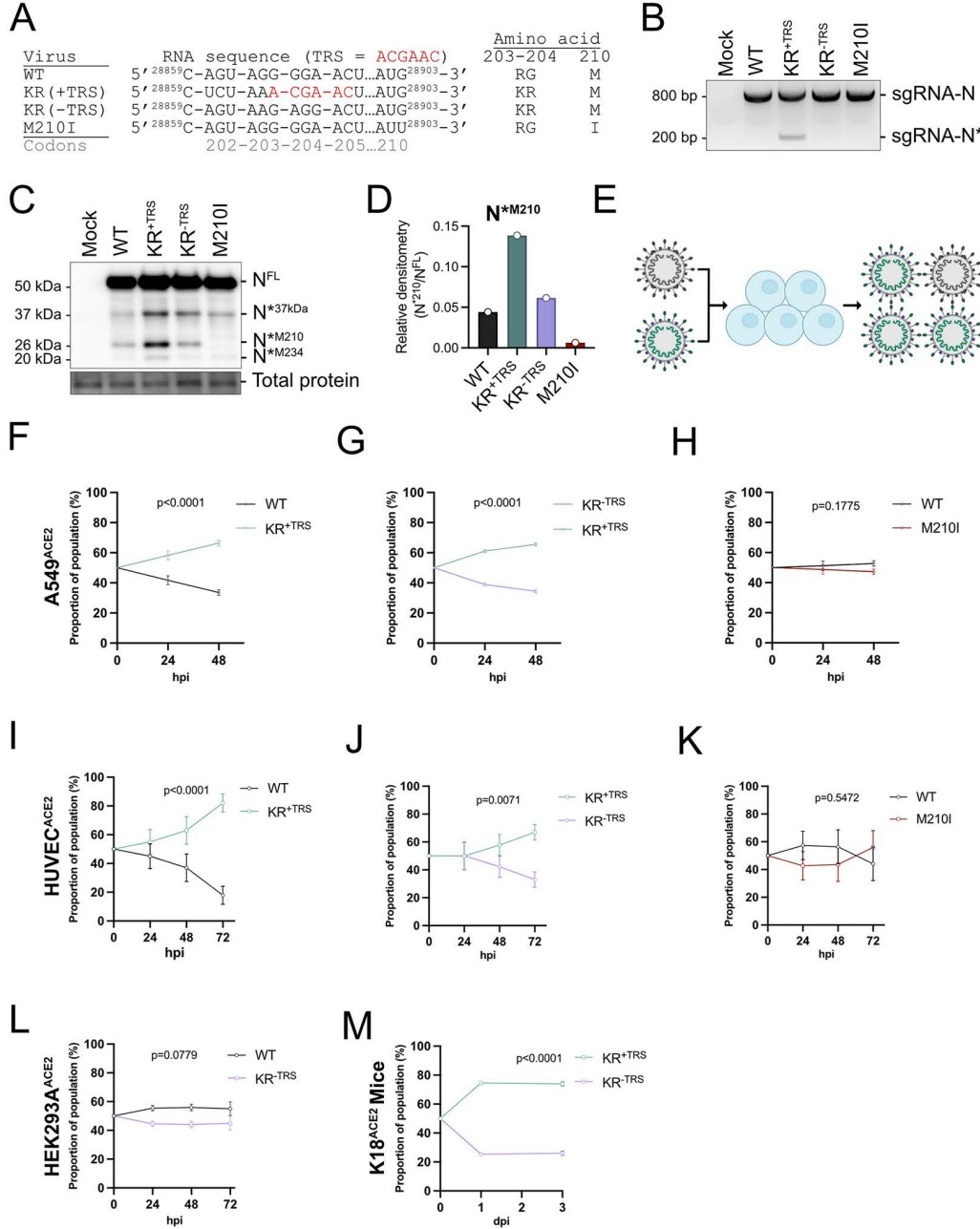

**Fig 3. Production of N\*M210 increases recombinant SARS-CoV-2 fitness. A.** Sequences of recombinant SARS-CoV-2 in the mutational hotspot in the N gene. Canonical TRS core sequence hexamer labeled in red. The sequences flanking the TRS in KR+TRS were based on the Gamma variant. Accession numbers - WT: PV935594.1, KR+TRS: PV952734, KR−TRS: PV936453, M210I: PV955084. **B.** sgRNA profile of recombinant SARS-CoV-2 was determined by infecting A549ACE2 cells (MOI = 4). RNA was harvested 24 hpi and subjected to RT-PCR and agarose gel electrophoresis. See S2 Supporting Data for full gel. **C, D.** N proteoform profile of SARS-CoV-2 variants was determined by infecting A549ACE2 cells (MOI = 4). Protein lysate was harvested 24 hpi and subjected to SDS-PAGE and immunoblotting with anti-N. Protein quantification was conducted by densitometry. N\*M210 proteoform abundance is presented relative to NFL abundance (N\*M210/NFL). See S2 Supporting Data for full blots and S1 Data for densitometry values. **E.** Schematic of co-infection competition assay made using Affinity Designer. **F–K.** A549ACE2 cells (F-H) or primary HUVECACE2 cells (I–K) were coinfected with equal infectious titers of the indicated recombinant virus to achieve a total MOI of 0.1 for A549ACE2 or MOI of 0.02 for HUVECACE2. Time 0 represents the inferred proportions of each recombinant virus based on infectious titer input. Intracellular RNA was harvested at the indicated times post-infection and subjected to probe-based RT-qPCR to differentiate recombinant virus abundance. $n = 3$ (E, F), $n = 6$ (G–I). A simple linear regression was conducted, standard error mean; SEM. See S1 Data for raw values. **L.** HEK293AACE2 cells were coinfected with equal infectious titers of WT or KR−TRS recombinant SARS-CoV-2 to achieve a total MOI of 0.01. Time 0 represents the inferred proportions of each recombinant based on infectious titer input. Intracellular RNA

was harvested at the indicated times post-infection and subjected to probe-based RT-qPCR to differentiate recombinant virus abundance. These data represent six independent biological replicates ($n = 6$). Statistics were performed using a simple linear regression, SEM. See S1 Data for raw values. **M.** 11-week-old K18$^{ACE2}$ mice (male and female) were intranasally inoculated with 2,500 TCID$_{50}$ of each recombinant virus. Day 0 represents the inferred proportions of each recombinant virus based on infectious titer input. On days 1 ($n = 4$) and 3 ($n = 5$) post-infection, mice were sacrificed and lungs were harvested for RNA. Viral species were differentiated using probe-based RT-qPCR. A simple linear regression was conducted, standard error mean; SEM. See S1 Data for raw values.

## N\*$^{M210}$ is a dsRNA-binding protein

The production of a truncated version(s) of a parent protein can significantly alter protein function by changing localization, post-translational modifications, or interactors [40–42]. We speculated that loss of the amino-terminal 209 amino acids of N$^{FL}$ could give N\*$^{M210}$ a unique function(s). N$^{FL}$ binds dsRNA, an event thought to be necessary to support viral RNA synthesis and antagonism of the antiviral response, both features that would promote efficient virus infection [33,34,43,44]. dsRNA-binding activity is dependent on two lysine residues, K257 and K261, in a dsRNA binding motif (dsRBM) [43,45] located in the CTD, a domain retained in N\*$^{M210}$ (Fig 2B). To test if N\*$^{M210}$ can bind dsRNA, lysates from cells expressing N$^{FL}$, N\*$^{M210}$, or an empty vector (EV) control were subjected to a pulldown using streptavidin beads conjugated to biotinylated poly I:C, a synthetic dsRNA mimic. As expected, N$^{FL}$ co-precipitated with dsRNA; however, N\*$^{M210}$ co-precipitated with dsRNA much more efficiently than N$^{FL}$ (Fig 4A and 4B). We mutated the dsRBM of N\*$^{M210}$ to generate N\*$^{M210-ΔdsRBM}$. No N\*$^{M210-ΔdsRBM}$ was detected in the eluate after dsRNA precipitation, indicating that the dsRBM is essential for dsRNA binding by N\*$^{M210}$.

To determine if N\*$^{M210}$ can interact with dsRNA in cells, we developed a poly I:C co-localization assay. After we confirmed that extranuclear Hoechst staining was a consequence of DNA transfection (S3A Fig), N$^{FL}$, N\*$^{M210}$, or EV-expressing cells were transfected with high molecular weight poly I:C and immunostained for dsRNA. Poly I:C transfection alone formed intracellular foci that contain dsRNA (Fig 4C, bottom left), consistent with the observations of others [2,46]. Poly I:C transfection also caused both N$^{FL}$ and N\*$^{M210}$ to redistribute; however, N$^{FL}$ and N\*$^{M210}$ redistributed into distinct foci. The majority of N$^{FL}$ was concentrated in perinuclear granules that were spatially distinct from dsRNA foci, while a minority of N$^{FL}$ co-localized with dsRNA-positive foci. By contrast, N\*$^{M210}$ was often found to co-localize with dsRNA foci, suggesting that N\*$^{M210}$ has superior dsRNA binding activity compared to N$^{FL}$ (Fig 4C). To quantify this, we determined the percent of total N$^{FL}$/N\*$^{M210}$ signal that co-localized with dsRNA. An average of 31.5 percent of the total N\*$^{M210}$ signal and 6.4 percent of N$^{FL}$ co-localized with dsRNA (Fig 4D), illustrating a differential enrichment of N\*$^{M210}$ in dsRNA-containing foci. When N\*$^{M210}$ lacked the dsRBM (N\*$^{M210-ΔdsRBM}$), the protein was not enriched in dsRNA foci (Fig 4C). We also confirmed that N\*$^{M210}$-dsRNA colocalization was not cell type dependent as we observed differential enrichment of N$^{FL}$ versus N\*$^{M210}$ in dsRNA-containing foci in A549 cells (S3B and S3C Fig). Furthermore, N\*$^{M210}$ readily sequestered low molecular weight poly I:C (S4A and S4B Fig). Poly I:C is an artificial dsRNA mimic; therefore, we validated the dsRNA colocalization assay using other types of dsRNA, including poly A:U which is reminiscent of AU-rich CoV RNAs [47]. We found that N\*$^{M210}$ relocalized in response to poly A:U dsRNA transfection, indicating that N\*$^{M210}$ sequestration is not sequence- or nucleotide-dependent (S4A and S4B Fig). Next, we wondered if N\*$^{M210}$ could interact with short dsRNAs. To test this, we transfected cells with a fluorescently labeled short ssRNA (18 bp) or a short sequenced-matched dsRNA (18 bp duplex) [48]. Like poly I:C and poly A:U, after transfection, the short ssRNAs and dsRNAs formed intracellular foci in the cytoplasm (Fig 4E). N$^{FL}$ failed to co-localize with ssRNA or dsRNA, whereas N\*$^{M210}$ strongly co-localized with the dsRNA, but not ssRNA (Fig 4E–4G). These data indicate that N\*$^{M210}$ preferentially interacts with dsRNA over ssRNA, including short duplexed RNAs.

## N\*$^{M210}$ interacts with N$^{FL}$ to increase N$^{FL}$ dsRNA sequestration capacity

During infection or overexpression, N$^{FL}$ forms homodimers via a dimerization motif in the CTD and higher-order oligomers via a leucine-rich helix in the linker [49–51]. Since N\*$^{M210}$ retains these motifs, we tested if N\*$^{M210}$ can interact with N$^{FL}$. We

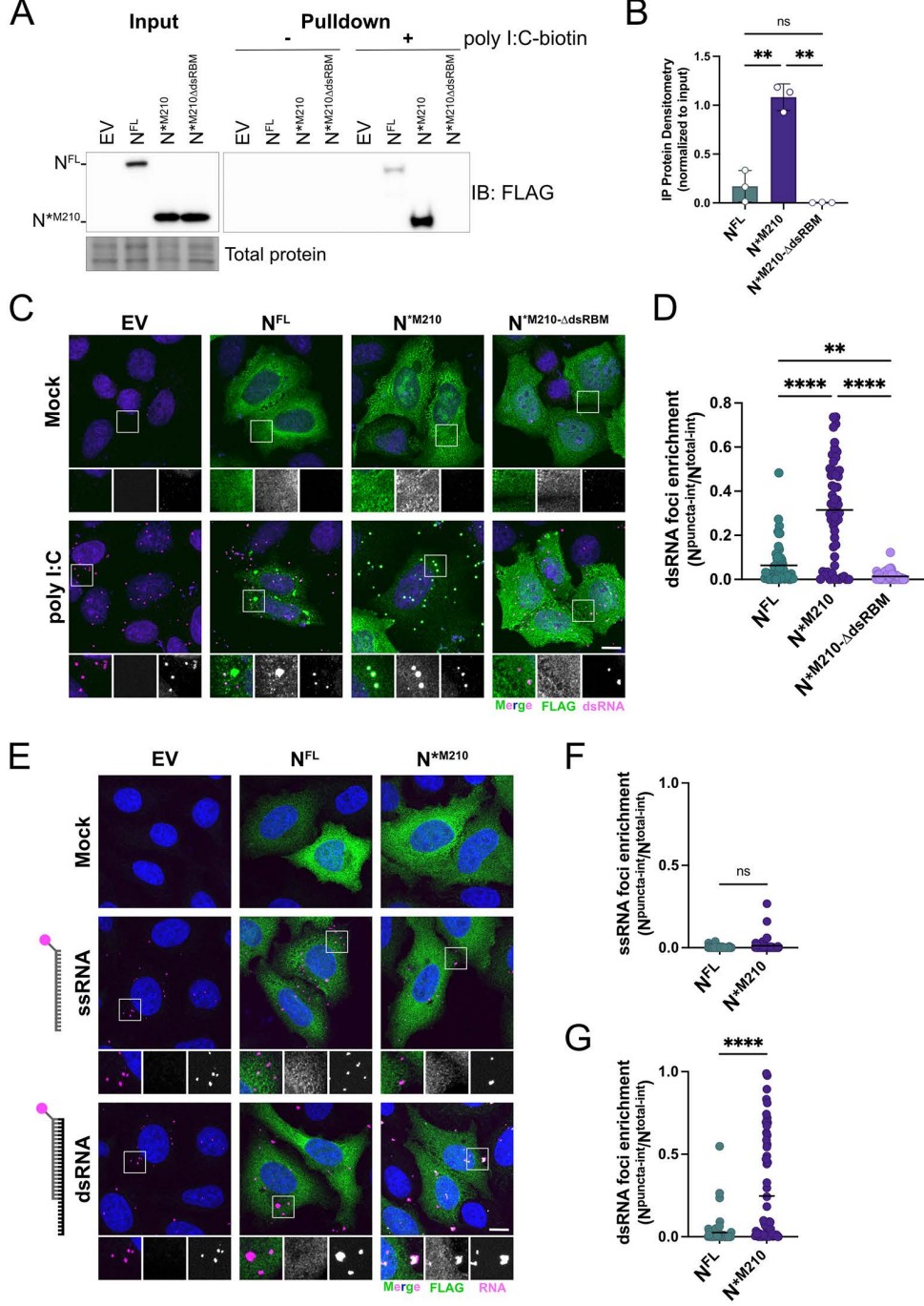

**Fig 4. N*^M210 has superior dsRNA binding abilities compared to N^FL. A, B.** EV, N^FL, N*^M210, or N*^M210-ΔdsRBM-transfected HEK293T cells were lysed and subjected to a pulldown using streptavidin beads conjugated to biotinylated poly I:C, or unconjugated beads. Eluted proteins were resolved by SDS-PAGE and immunoblotting. Protein abundance after pulldown was enumerated by densitometry and normalized to each respective input control. These data represent three independent biological replicates ($n = 3$). Statistics were performed using a one-way ANOVA with Dunnett's post-hoc analysis, (** $p < 0.0021$), mean; standard deviation. See S3 Supporting Data for full blots and S1 Data for densitometry values. **C.** HeLa cells were transfected with FLAG-tagged N^FL, N*^M210, or N*^M210-ΔdsRBM, or an empty vector (EV) control. Internal methionine residues were mutated to ensure that only the indicated proteoform of N was expressed ([N^FL; M210I and M234V], [N*^M210 and N*^M210-ΔdsRBM; M234V]). Twenty-four hours post-transfection, cells were transfected with 0.5 μg high molecular weight poly I:C or mock-transfected. Three hours after poly I:C treatment, cells were fixed and immunostained with the J2 antibody (dsRNA; Alexa 647) and FLAG antibody (N proteoform; Alexa 488). Nuclei were stained with Hoechst. A maximum intensity projection is presented here. One representative experiment of three independent replicates is shown. Scale bar = 10 μm. **D.** Enrichment of N proteoforms with dsRNA

was determined using CellProfiler using confocal images. Enrichment was measured by determining the proportion of $N^{FL}/N^{*M210}$ that is cytoplasmic vs. co-localized with dsRNA foci. The mean integrated intensity of the N proteoform overlapping with dsRNA foci was divided by the mean integrated intensity of the respective N proteoform in the entire cytoplasm. Each data point represents a single cell. These data represent three independent biological replicates ($n = 3$) with >18 cells measured per condition, per replicate. Statistics were performed using a Kruskal–Wallis $H$ test with Dunn's correction (** $p < 0.0021$, **** $p < 0.0001$), mean. See S1 Data for quantification values. **E.** HeLa cells were transfected with FLAG-tagged $N^{FL}$, $N^{*M210}$ or EV as in A. Twenty-four hours post-transfection, cells were transfected with 0.9 μg of fluorescein-labeled (18 bp) ssRNA or dsRNA probes (18 bp duplex, 20 bp overhang). Three hours later, cells were fixed and immunostained with the FLAG antibody (N proteoform; Alexa 647). Nuclei were stained with Hoechst. A maximum intensity projection is presented here. One representative experiment of three independent replicates is shown. Scale bar = 10 μm. **F, G.** Enrichment of N proteoforms with RNA probes as in D. These data represent three independent biological replicates ($n = 3$) with 20 cells measured per condition, per replicate. Statistics were performed using a Mann-Whitney test (**** $p < 0.0001$), mean. See S1 Data for quantification values.

co-expressed $N^{*M210}$-FLAG and $N^{FL-WT}$-HA in HEK293T cells and conducted co-immunoprecipitation assays. We found that $N^{FL}$ co-precipitated with $N^{*M210}$ (Fig 5A). The expression construct for $N^{FL-WT}$ contains residues M210 and M234 and produces low levels of $N^{*M210}$ and $N^{*M234}$ in addition to $N^{FL}$ (S1F Fig). Because these N* proteoforms also co-precipitated with $N^{*M210}$, these data reveal that $N^{*M210}$ can interact with $N^{*M234}$ and $N^{*M210}$ proteins, forming homo- and heterodimers (Fig 5B). Co-precipitation of all dimeric combinations ($N^{FL}:N^{*M210}$, $N^{*M210}:N^{*M210}$, and $N^{*M210}:N^{*M234}$) was substantially reduced upon RNase A treatment, suggesting that these interactions are enhanced by the presence of RNA (Fig 5A).

During an infection, $N^{FL}$ and $N^{*M210}$ are co-expressed in the same cell at the same time (S1G Fig) and given that $N^{*M210}$ can interact with $N^{FL}$, we wondered if $N^{*M210}$ can alter the behavior of $N^{FL}$. In other words, does an $N^{FL}:N^{*M210}$ dimer have enhanced dsRNA co-localization compared to an $N^{FL}:N^{FL}$ dimer? To test this, we co-expressed $N^{FL}$-HA and $N^{*M210}$-FLAG, then transfected these cells with poly I:C. We found that cells expressing only $N^{FL}$ formed puncta that were distinct from dsRNA-positive foci after poly I:C transfection as in Fig 4C (Fig 5C). By contrast, after co-transfection of $N^{FL}$ and $N^{*M210}$, $N^{FL}$ often co-localized with dsRNA (Fig 5C). During infection, $N^{FL}$ is produced ~5 times more than $N^{*M210}$ (Fig 2G and 2I). To understand the relative amount of $N^{*M210}$ required to re-localize $N^{FL}$ to dsRNA-positive foci, we altered the ratio of $N^{FL}/N^{*M210}$ expression constructs delivered by transfection. We found that co-expression of $N^{*M210}$ increased the propensity for $N^{FL}$ to co-localize with dsRNA foci in a dose-dependent manner. Transfection of an 8:1 ratio of $N^{FL}/N^{*M210}$ plasmid DNA was sufficient to increase $N^{FL}$ co-localization with dsRNA; however, $N^{FL}$-dsRNA co-localization was further enhanced after transfection of plasmid DNA using a 2:1 ratio of $N^{FL}/N^{*M210}$ (Fig 5D–5F). These data show that $N^{*M210}$ and $N^{FL}$ interact, and that this event alters the ability of $N^{FL}$ to co-localize with dsRNA.

## $N^{*M210}$ is sufficient to block dsRNA-induced antiviral immune responses

The presence of dsRNA is a telltale sign of a viral infection. As such, dsRNA is a potent inducer of various antiviral immune responses [52,53]. There are three primary dsRNA-induced response axes: (i) Protein kinase R (PKR) binds dsRNA, auto-phosphorylates, then phosphorylates eukaryotic initiation factor 2α (eIF2α), which triggers translational arrest and SG formation. (ii) RIG-I-like receptors (RLRs) bind dsRNA causing phosphorylation of interferon regulatory factor-3 (IRF3), which induces interferon transcription. (iii) 2′-5′-oligoadenylate synthetase (OAS) binds dsRNA, activating RNase L, which promotes RNA decay and translational arrest [52]. $N^{FL}$ has been shown to block all three of these pathways [29,43]. To test if $N^{*M210}$ is also able to dampen dsRNA-induced immune responses, A549 cells expressing $N^{FL}$, $N^{*M210}$, or EV were transfected with poly I:C. Immunoblotting revealed that both $N^{FL}$ and $N^{*M210}$ were sufficient to dampen IRF3 and PKR phosphorylation (Fig 6A–6C). Furthermore, $N^{FL}$ and $N^{*M210}$ were sufficient to block *IFNβ* mRNA induction (Fig 6D) and the OAS-RNase L axis, using rRNA decay as a proxy for RNase L activation (Fig 6I). The blockade of these three dsRNA-induced responses required the dsRBM of $N^{*M210}$ (Fig 6E–6H and 6J), suggesting that $N^{*M210}$ antagonizes these pathways by sequestering dsRNA (Fig 6K). Despite the superior ability of $N^{*M210}$ to bind and co-localize with dsRNA relative to $N^{FL}$, in these assays we did not observe a significant difference in the ability of these two constructs to block dsRNA-induced responses.

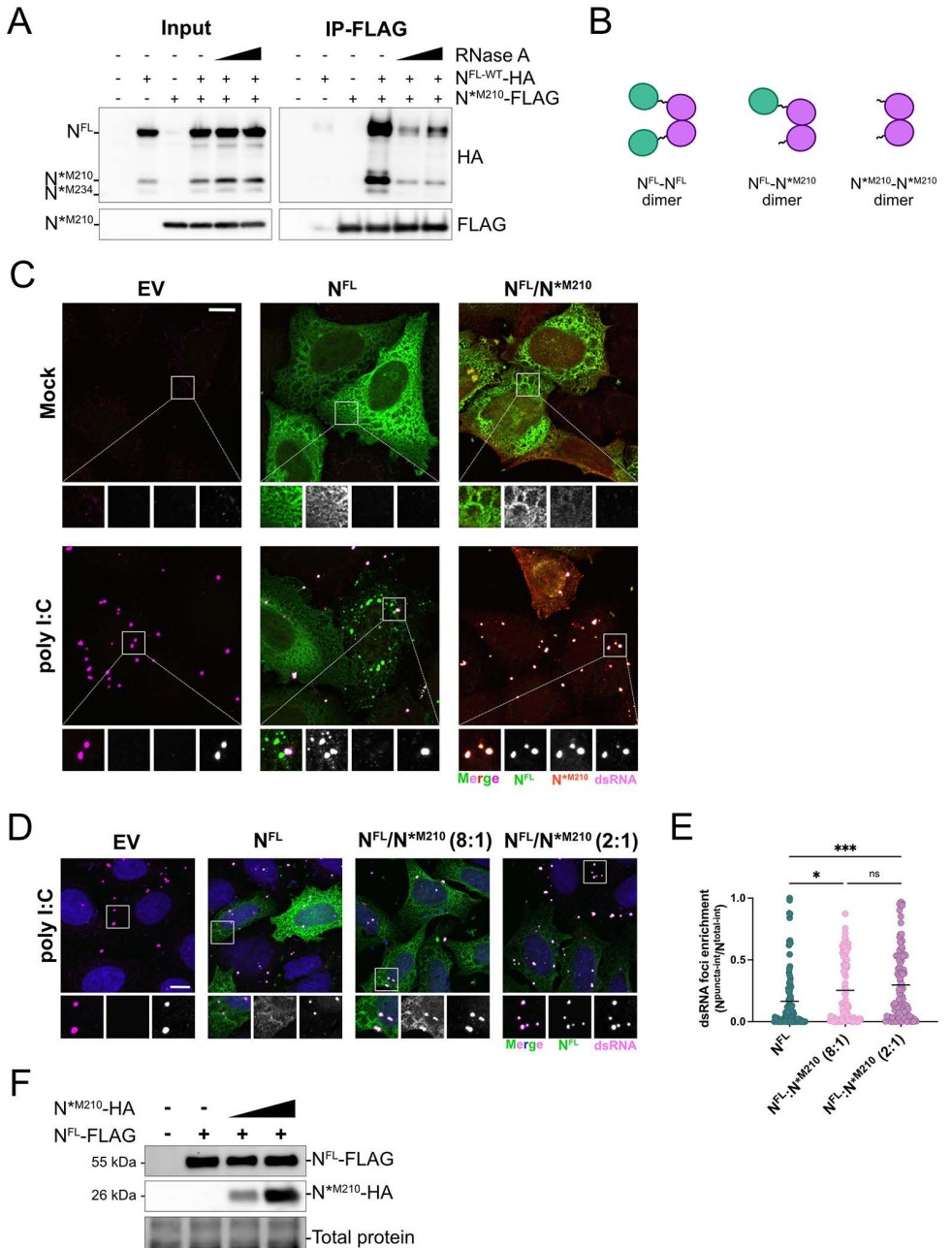

**Fig 5. N\*^M210 interacts with N^FL to increase N^FL co-localization with dsRNA. A.** HEK293T cells were co-transfected with HA-tagged N^FL-WT, with intact downstream methionines (M210 and M234), and FLAG-tagged N\*^M210. Twenty-four hours post-transfection cells were lysed, treated with 10 or 50 μg/mL RNase A, and incubated with anti-FLAG antibody overnight at 4 °C. Immunoprecipitation was performed using magnetic Dynabeads and samples were resolved by SDS-PAGE and immunoblotted with anti-FLAG and anti-HA antibodies. See S4 Supporting Data for full blots. **B.** Schematic representation (Affinity Designer) of N proteoform homo- and heterodimers based on A. **C.** HeLa cells expressing N^FL-FLAG alone or co-expressing N^FL-FLAG and N\*^M210-HA were transfected with 0.5 μg poly I:C. Three hours post-transfection, cells were fixed and immunostained with the FLAG antibody (N^FL; Alexa 488), the HA antibody (N\*^M210; Alexa 405), and the J2 antibody (dsRNA; Alexa 555). A maximum intensity projection (MIP) is presented here. Scale bar = 10 μm. **D.** HeLa cells expressing N^FL-FLAG alone or co-expressing N^FL-FLAG and N\*^M210-HA at an 8:1 or 2:1 N^FL:N\*^M210 ratio were transfected with 0.5 μg poly I:C. Three hours post-transfection, cells were fixed and immunostained with the FLAG antibody (N^FL; Alexa 488) and the J2 antibody (dsRNA; Alexa 647). Nuclei were stained with Hoechst. A maximum intensity projection (MIP) is presented here. Scale bar = 10 μm. **E.** Enrichment of N^FL with dsRNA from D. was calculated as in Fig 4D. These data represent three independent biological replicates (*n* = 3) with 37 cells measured per condition, per replicate. Statistics were performed using a Kruskal–Wallis *H* test with Dunn's correction (**** *p* < 0.0001), mean. See S1 Data for quantification values. **F.** Protein lysate reserved from D. was resolved by SDS-PAGE and immunoblotting to validate protein expression. See S4 Supporting Data for full blots.

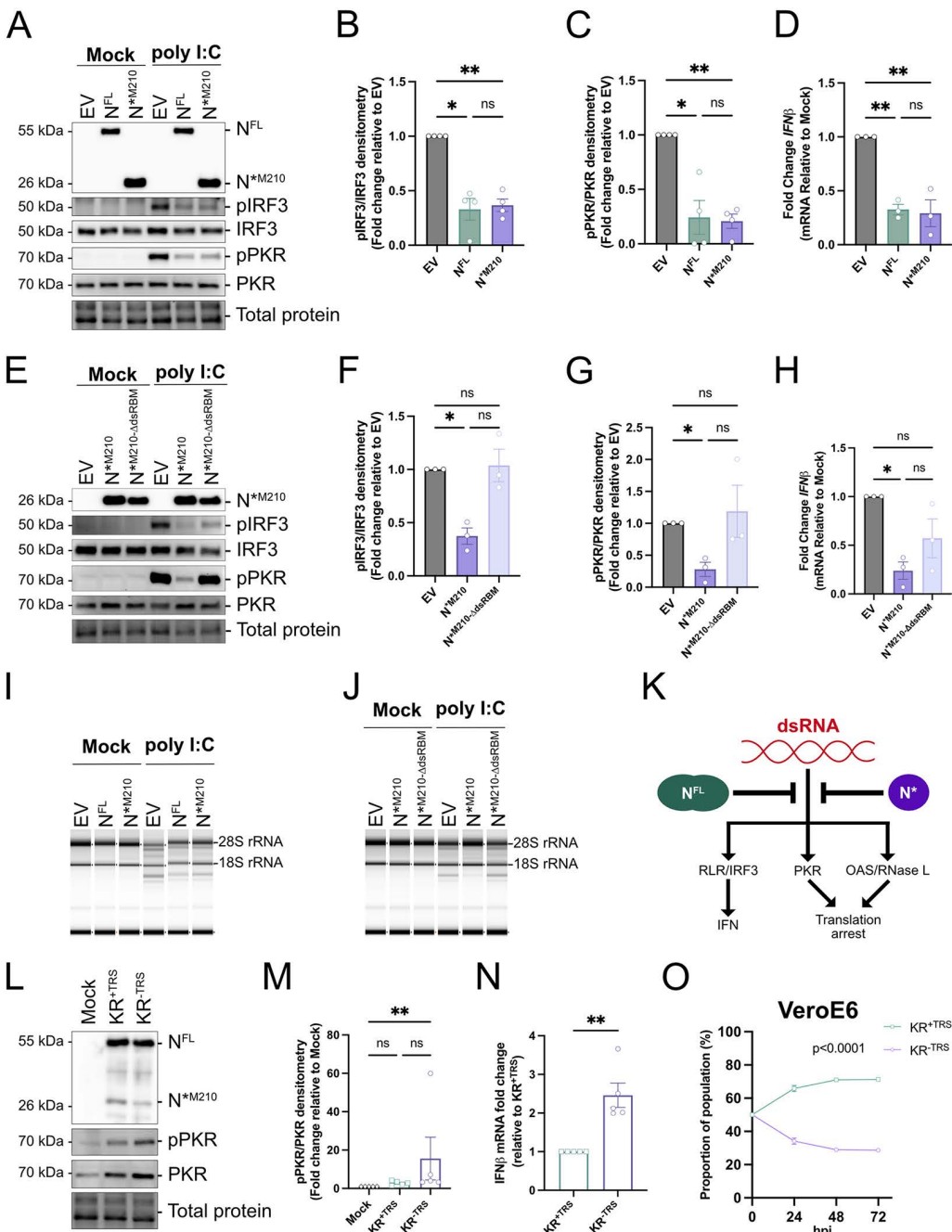

**Fig 6. N*^M210 is sufficient to block cellular dsRNA immune responses. A.** A549 cells were transduced with recombinant lentiviruses to ectopically express N proteoforms or EV. Internal methionine residues were mutated to ensure that only the indicated proteoform of N was expressed ([N^FL; M210I and M234V], [N*^M210; M234V]). Ninety-six hours post-transduction, cells were transfected with 1.0 µg high molecular weight poly I:C or mock-transfected. Three hours post-treatment, protein lysate was harvested and resolved by SDS-PAGE and immunoblotted with antibodies specific for FLAG (N proteoforms), pIRF3, IRF3, pPKR, and PKR. A representative experiment is shown. See S5 Supporting Data for full blots. **B, C.** Protein levels from poly I:C-treated samples in A. were quantified by densitometry in ImageLab. pIRF3 and pPKR abundances were measured relative to total IRF and PKR, respectively, and normalized to EV. These data represent four independent biological replicates ($n = 4$). Statistics were performed using a one-way ANOVA with Dunnett's post-hoc analysis, with standard error mean (SEM). See S1 Data for densitometry values. **D.** N proteoforms were ectopically expressed in A549 cells as in A. Cells were treated with 1.0 µg high molecular weight poly I:C or mock-transfected. Three hours post-poly I:C treatment, intracellular RNA was harvested and *IFN*β RNA abundance was measured by RT-qPCR. These data are represented as fold-change relative to the no poly I:C treatment from three independent biological replicates ($n = 3$). Statistics were performed using a one-way ANOVA with Dunnett's post-hoc

analysis with standard error mean (SEM). See S1 Data for quantification values. **E.** N proteoforms were ectopically expressed, treated with poly I:C, and subjected to SDS-PAGE and immunoblotting as in A. One of three independent experiments shown. See S5 Supporting Data for full blots. **F, G.** Protein levels from poly I:C-treated samples in E. were quantified by densitometry as in B and C. These data represent three independent biological replicates ($n = 3$). Statistics were performed using a one-way ANOVA with Dunnett's post-hoc analysis, with standard error mean (SEM). See S1 Data for densitometry values. **H.** N proteoforms were ectopically expressed in A549 cells as in E and treated with poly I:C. Three hours post-poly I:C treatment, intracellular RNA was harvested and *IFNβ* RNA abundance was measured by RT-qPCR. These data are represented as fold-change relative to the no poly I:C treatment from three independent biological replicates ($n = 3$). Statistics were performed using a one-way ANOVA with Dunnett's post-hoc analysis, with standard error mean (SEM). See S1 Data for quantification values. **I, J.** RNA lysates from D and H were subjected to automated electrophoresis using an Agilent 4200 TapeStation system to assess rRNA integrity. One representative of three independent experiments shown ($n = 3$). See S5 Supporting Data for full digital gels. **K.** Schematic representation (Affinity Designer) of N$^{FL}$/N*$^{M210}$ inhibition of dsRNA-induced responses. **L.** Primary HUVEC$^{ACE2}$ cells were infected with KR$^{+TRS}$ or KR$^{-TRS}$ recombinant SARS-CoV-2 viruses (MOI = 4), or mock-infected. Twenty-four hours post-infection, protein lysate was harvested and resolved by SDS-PAGE and immunoblotted with antibodies specific for N, pPKR, and PKR. A representative experiment is shown. See S5 Supporting Data for full blots. **M.** Protein levels from rSARS-CoV-2-infected cells from L. were quantified as in B. PKR levels and relative pPKR abundances were measured and normalized to EV. These data are represented as fold-change relative to mock from five independent biological replicates ($n = 5$). Statistics were performed using a Friedman test (non-normal deviation), (** $p < 0.0021$, SEM). See S1 Data for densitometry values. **N.** HUVEC$^{ACE2}$ cells were infected with KR$^{+TRS}$ or KR$^{-TRS}$, as in L. Twenty-four hours post-infection, intracellular RNA was harvested and *IFNβ* RNA abundance was measured by RT-qPCR. Data are represented as fold-change relative to KR$^{+TRS}$. These data represent five independent biological replicates ($n = 5$). Statistics were performed using a ratio paired *T* test, (** $p < 0.0021$, SEM). See S1 Data for quantification values. **O.** VeroE6 cells were coinfected with equal infectious titers of KR$^{+TRS}$ and KR$^{-TRS}$ recombinant SARS-CoV-2 viruses to achieve a total MOI of 0.01. Time 0 represents the inferred proportions of each recombinant based on infectious titer input. Intracellular RNA was harvested at the indicated times post-infection and subjected to probe-based RT-qPCR to differentiate recombinant virus abundance. These data represent three independent biological replicates ($n = 3$). Statistics were performed using a simple linear regression, SEM. See S1 Data for quantification values.

To determine if N*$^{M210}$ influences dsRNA-induced responses in the context of an infection, we infected primary HUVEC$^{ACE2}$ cells with KR$^{+TRS}$ or KR$^{-TRS}$ recombinant viruses. Infection with both KR$^{+TRS}$ and KR$^{-TRS}$ induced PKR activation (pPKR/PKR) as evidenced by increased abundance of phosphorylated PKR after immunoblotting (Fig 6L). However, only infection with KR$^{-TRS}$, not KR$^{+TRS}$ infection, significantly increased phospho-PKR levels relative to mock conditions (Fig 6M), suggesting that KR$^{+TRS}$ displays a superior ability to supress PKR activation during infection. Furthermore, KR$^{+TRS}$-infected cells consistently produced ~2-fold less *IFNβ* mRNA than KR$^{-TRS}$-infected cells, highlighting the potent effect of enhanced N*$^{M210}$ production on immune evasion during an authentic viral infection (Fig 6N). Because viruses with greater N*$^{M210}$ abundance show reduced IFN induction, we wondered if enhanced IFN suppression was the mechanism underlying the fitness advantage displayed by KR$^{+TRS}$ during co-infection (Fig 3G and 3J). To test this, we conducted co-infection competition assays in VeroE6 cells, which are unable to produce type I IFNs [54]. We reasoned that if the replication advantage of KR$^{+TRS}$ was exclusively due to a superior ability to block IFN, KR$^{+TRS}$ and KR$^{-TRS}$ would have equal fitness in VeroE6 cells. However, contrary to the hypothesis, we observed that even in IFN-deficient cells, KR$^{+TRS}$ outcompeted KR$^{-TRS}$, suggesting that the fitness difference between KR$^{+TRS}$ and KR$^{-TRS}$ is not solely from N*$^{M210}$-mediated IFN suppression (Fig 6O). There are two important interpretations from these data: (i) N*$^{M210}$ potently antagonizes dsRNA-induced antiviral responses, likely via dsRNA sequestration, and (ii) N*$^{M210}$ promotes viral replication using a mechanism that is independent of IFN responses.

### N*$^{M210}$ inhibits cellular ribonucleoprotein granules via the dsRBM

Ras GTPase-activating RNA-binding protein 1 (G3BP1) is an RNA-binding protein that can form ribonucleoprotein granules in response to cellular stress or viral infection [5,8,15,55]. G3BP1-positive foci include SGs and RLBs. SGs form due to translation initiation blockage, such as eIF2α phosphorylation by stress-responsive kinases like PKR or heme-regulated inhibitor (HRI) [5,56,57], while RLBs form following RNase L activation by dsRNA [58]. The precise role of G3BP1 during coronaviral infection is complex; however, G3BP1 protein and G3BP1-containing granules have been reported to be antiviral, providing an obstacle to productive viral infection [8,11,59]. SARS-CoV-2 N$^{FL}$ has been shown to inhibit G3BP1 foci formation by directly interacting with G3BP1 via the φxF motif in the amino-terminus of N$^{FL}$ [31,43,60–62]. N*$^{M210}$ lacks

the φxF motif, so we hypothesized that N*$^{M210}$ would be unable to antagonize G3BP1 foci. To test this, cells expressing N*$^{M210}$, N$^{FL}$, or EV were treated with poly I:C, which should induce RLB formation. We found that both N$^{FL}$ and N$^{FL-ΔdsRBM}$ readily localized with G3BP1. N*$^{M210}$ and N*$^{M210-ΔdsRBM}$ did not localize with G3BP1 (Fig 7A), likely because N*$^{M210}$ does not retain the G3BP1-interacting motif (Fig 2B). Instead, following poly I:C treatment, N*$^{M210}$ formed G3BP1-negative puncta, whereas N*$^{M210-ΔdsRBM}$ remained diffuse. Consistent with the literature, we found that N$^{FL}$ reduced G3BP1 foci formation, but this effect was marginal, especially relative to N*$^{M210}$ which potently blocked G3BP1 foci using a mechanism dependent on the dsRBM (Fig 7A and 7B). By co-staining for dsRNA, FLAG (N$^{FL}$/N*$^{M210}$), and TIAR (an alternative SG/RLB marker) in the same experiment (Fig 7C), we confirmed that poly I:C transfection induced differential localization of N$^{FL}$ and N*$^{M210}$; N$^{FL}$ primarily localized in RLBs whereas N*$^{M210}$ localized to dsRNA-positive foci (Fig 7D–7G). Unlike N$^{FL}$, N*$^{M210}$ did not co-immunoprecipitate with G3BP1 even after poly I:C transfection (Fig 7H). Collectively, these data suggest that N*$^{M210}$ strongly blocks RLB formation using a mechanism that is independent of G3BP1 binding, but dependent on the dsRBM.

Given that N*$^{M210}$ inhibited RNase L (Fig 6), we predicted that N*$^{M210}$ inhibited RLB formation by sequestering the inducer, dsRNA. If so, N*$^{M210}$ should only be able to inhibit G3BP1 foci induced by exogenous addition of dsRNA and should not block foci induced by alternative stressors. To test this, we induced G3BP1 foci with sodium arsenite (SA), which induces SG formation by activating HRI [63]. We observed that both N$^{FL}$ and N$^{FL-ΔdsRBM}$ co-localized with G3BP1 in SGs, consistent with both proteins retaining the φxF motif, whereas N*$^{M210}$ and N*$^{M210-ΔdsRBM}$ remained diffuse in the cytoplasm (Fig 7I and 7J). N$^{FL}$ blocked SG formation following SA treatment using a mechanism largely dependent on the dsRBM, as N$^{FL-ΔdsRBM}$ did not significantly reduce SGs. Contrary to our expectation, SA-induced SGs were strongly inhibited by N*$^{M210}$, and this effect also required the dsRBM, as N*$^{M210-ΔdsRBM}$ failed to inhibit SGs (Fig 7I and 7J). There are two important interpretations from these data: (i) N*$^{M210}$ does not only block dsRNA-induced G3BP1 foci by sequestering dsRNA, but instead limits G3BP1 foci formation from various inducers and (ii) the dsRNA-binding motif of N*$^{M210}$ is required to prevent G3BP1 foci formation, even when induced via a dsRNA-independent pathway.

In addition to blocking RLBs and SGs, N$^{FL}$ also induces processing body (P-body) disassembly [14]. P-bodies are also membraneless granules, but unlike RLBs and SGs, P-bodies do not contain G3BP1 and are constitutively present in the cytoplasm where they act as sites of RNA repression and decay [64–67]. P-bodies are hypothesized to have antiviral activity; therefore, many viruses, including SARS-CoV-2, trigger P-body disassembly during infection [4,9,10,14]. Unlike G3BP1 foci, P-bodies are not induced by dsRNA [58]. To test if N*$^{M210}$ is sufficient to disassemble P-bodies, HUVEC cells expressing N$^{FL}$ or N*$^{M210}$ with or without the dsRBM were fixed and immunostained (S5A Fig). P-bodies were quantified by measuring Hedls, a resident P-body protein. Both N$^{FL}$ and N*$^{M210}$ decreased P-body numbers and both proteoforms required an intact dsRBM to mediate P-body disassembly (S5B Fig). Collectively, these data show that N*$^{M210}$ potently blocks P-bodies and G3BP1 foci, including RLBs and SGs, using a mechanism that relies on its dsRBM, suggesting that the RNA-binding ability of N*$^{M210}$ broadly inhibits ribonucleoprotein granule formation.

Overexpression of N*$^{M210}$ potently inhibited G3BP1 foci, so we wondered if this remained true in the context of infection. In other words, would a virus with increased N*$^{M210}$ production more readily inhibit SG/RLB formation? To test this, we infected VeroE6 cells with rSARS-CoV-2 KR$^{+TRS}$ (high N*$^{M210}$) or rSARS-CoV-2 KR$^{-TRS}$ (low N*$^{M210}$) and labeled SGs/RLBs using two different markers (G3BP1 and TIAR). Previous work has shown that during infection, high concentrations of N$^{FL}$ inhibit SG and RLB formation; therefore, we harvested cells at 6 hpi, an infection time when N$^{FL}$ levels are expected to be lower [15,58]. Immunofluorescent staining of these infected revealed heterogenous expression of N, where ~80 percent of infected cells were "high N expressors" and displayed N staining diffusely localized throughout the cytoplasm, and ~20 percent of infected cells were 'low N expressors' and displayed punctate N staining (Fig 8A; 17% of KR$^{+TRS}$-infected cells and 25% of KR$^{-TRS}$-infected cells). These cells were also immunostained for G3BP1 and TIAR. Infection with both viruses induced small G3BP1 and TIAR foci, likely representing small SGs or RLBs (Fig 8B–8E) and consistent with recent work by Long and colleagues (2024) showing that G3BP1 foci form at early infection times but are inhibited at later infection stages [15]. We observed the presence of G3BP1 and TIAR foci more often in cells that were 'low N expressors',

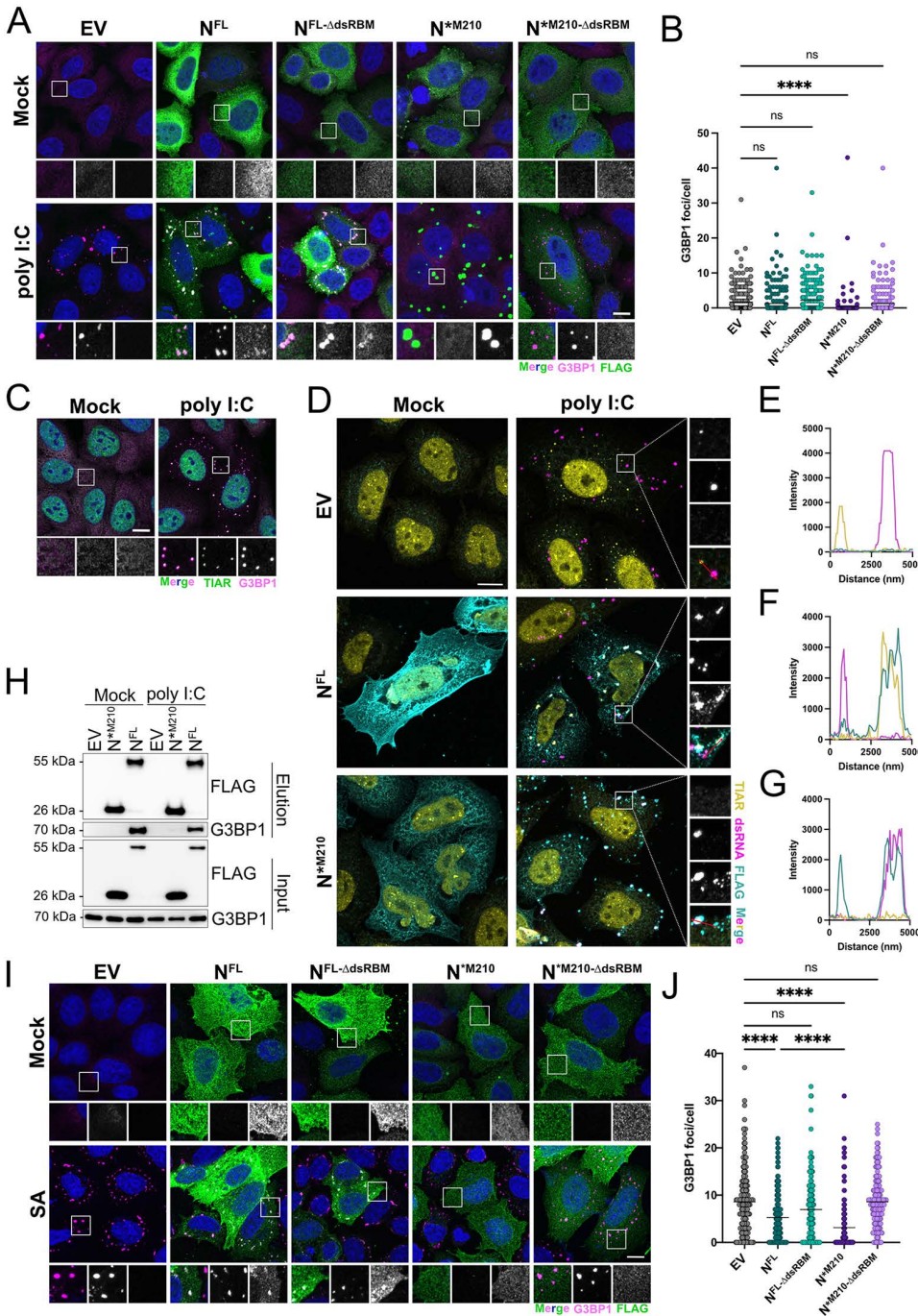

**Fig 7. The dsRNA binding domain of N\*M210 is required to block dsRNA-induced and arsenite-induced G3BP1 foci. A.** HeLa cells were transfected with FLAG-tagged N^FL, N^FL-ΔdsRBM, N\*M210, or N\*M210-ΔdsRBM, or an empty vector (EV) control. Internal methionine residues were mutated to ensure only the indicated proteoform of N was expressed ([N^FL and N^FL-ΔdsRBM; M210I and M234V], [N\*M210 and N\*M210-ΔdsRBM; M234V]). Twenty-four hours post-transfection, cells were transfected with 0.5 μg poly I:C or mock-transfected. Three hours after poly I:C treatment, cells were fixed and immunostained with a G3BP1 antibody (RLBs; Alexa 647) and FLAG antibody (N proteoform; Alexa 488). Nuclei were stained with Hoechst. A maximum intensity projection is presented here. Scale bar = 10 μm. **B.** RLBs were quantified using CellProfiler by measuring G3BP1 puncta in N-expressing cells (thresholded by FLAG staining) or EV-transfected cells. These data represent three independent biological replicates (*n* = 3) with >50 cells measured per condition, per replicate. Each datapoint represents a single cell. Statistics were performed using a Kruskal–Wallis *H* test with Dunn's correction (**** *p* < 0.0001, mean). See S1 Data for quantification values. **C.** HeLa cells were transfected with 0.5 μg poly I:C. Three hours after poly I:C treatment, cells

were fixed and immunostained with antibodies specific for TIAR (Alexa 488) and G3BP1 (Alexa 647). Nuclei were stained with Hoechst. A maximum intensity projection (MIP) is presented here. Scale bar = 10 μm. **D.** HeLa cells were transfected with $N^{FL}$, $N^{*M210}$, or EV, and treated with 0.5 μg poly I:C as in **A**. Three hours after poly I:C treatment, cells were fixed and immunostained with antibodies specific for TIAR (RLBs; Alexa 405), J2 (dsRNA; Alexa 647), and FLAG (N proteoform; Alexa 488). A maximum intensity projection (MIP) is presented here. Scale bar = 10 μm. **E–G.** Intensity histograms were obtained in Zen Blue. Line was drawn through confocal images from D (inset, red line) and intensity maps are depicted. See S1 Data for quantification values. **H.** HEK293T cells were transfected with pcDNA-$N^{FL}$-FLAG (M210I, M234V), pcDNA-$N^{*M210}$-FLAG (M234V), or an empty vector (EV) control. Forty-eight hours post-transfection, cells were transfected with 10 μg of high molecular weight poly I:C. Three hours post-poly I:C treatment, cells were lysed and incubated with anti-FLAG antibody overnight at 4 °C. Immunoprecipitation was performed using magnetic Dynabeads (Thermo-Fisher) and samples were resolved by SDS-PAGE and immunoblotted with anti-FLAG antibody and anti-G3BP1 antibody. One of three independent experiments shown. See S6 Supporting Data for full blots. **I.** HeLa cells were transfected with FLAG-tagged $N^{FL}$, $N^{FL-\Delta dsRBM}$, $N^{*M210}$, or $N^{*M210-\Delta dsRBM}$, or an empty vector (EV) control. Internal methionine residues were mutated to ensure that only the indicated proteoform of N was expressed ([$N^{FL}$ and $N^{FL-\Delta dsRBM}$; M210I and M234V], [$N^{*M210}$ and $N^{*M210-\Delta dsRBM}$; M234V]). Twenty-four hours post-transfection, cells were treated with 500 μM sodium arsenite for 1 hour before the cells were fixed and immunostained with a G3BP1 antibody (SGs; Alexa 647) and FLAG antibody (N proteoform; Alexa 488). Nuclei were stained with Hoechst. A maximum intensity projection is presented here. Scale bar = 10 μm. **J.** G3BP1 foci were quantified as in Fig 7B. These data represent four independent biological replicates (*n* = 4) with 40 cells measured per condition, per replicate. Each datapoint represents a single cell. Statistics were performed using a Kruskal–Wallis *H* test with Dunn's correction (**** $p < 0.0001$, mean). See S1 Data for quantification values.

suggesting that, as $N^{FL}$ accumulates, SGs/RLBs are more likely to be inhibited (Figs 8B–8E, S5C, and S5D). Indeed, quantification of G3BP1 foci and TIAR foci in all N-positive cells (low and high expressors) shows that both $KR^{+TRS}$ and $KR^{-TRS}$ were sufficient to inhibit foci formation (S5C and S5D Fig). However, in cells where $N^{FL}$ is limiting (low expressors), $KR^{+TRS}$ was more effective at inhibiting G3BP1 and TIAR foci formation than $KR^{-TRS}$ (Fig 8B–8E). There are two important implications from this data: (i) at early SARS-CoV-2 infection times, foci reminiscent of RLBs or SGs form and (ii) when $N^{FL}$ is limiting, viruses that make more $N^{*M210}$ are superior at blocking G3BP1/TIAR foci formation.

Various studies suggest that G3BP1 foci elicit antiviral activity [7,8,11,57,68,69]. If G3BP1 foci pose a hurdle for a successful virus infection and $N^{*M210}$ blocks the formation of these granules, we wondered if the difference in virus fitness between $KR^{+TRS}$ and $KR^{-TRS}$ could be explained by the ability of $N^{*M210}$ to block G3BP1 foci formation. To directly test if $KR^{+TRS}$ outcompetes $KR^{-TRS}$ because of an enhanced ability to inhibit G3BP1 foci formation, we conducted co-infection competition assays in either WT or G3BP1 knockout (ΔG3BP1) HEK293A$^{ACE2}$ cells (S5E Fig) [70]. Consistent with A549$^{ACE2}$ and HUVEC$^{ACE2}$ cells, $KR^{+TRS}$ readily outcompeted $KR^{-TRS}$ in WT HEK293A$^{ACE2}$ cells, as by 72 hpi $KR^{+TRS}$ comprised 80 percent of the virus population (Fig 8F). In ΔG3BP1 HEK293A$^{ACE2}$ cells, $KR^{+TRS}$ still outcompeted $KR^{-TRS}$; however, the fitness advantage of $KR^{+TRS}$ was diminished in knockout cells compared to WT cells, with $KR^{+TRS}$ plateauing at 60 percent dominance at 72 hpi (Fig 8G and 8H). G3BP1 and its paralog G3BP2 are required for SG formation [71]. To determine if G3BP1/2 dual knockout influences the virus fitness landscape, we repeated co-infection competition assays in WT or ΔG3BP1/2 A549$^{ACE2/TMPRSS2}$ cells. We confirmed the fitness of $KR^{+TRS}$ relative to $KR^{-TRS}$ in WT cells, with 75 percent of viral sequences corresponding to $KR^{+TRS}$ by 72 hpi (Fig 8I). However, this fitness advantage was lost in ΔG3BP1/2 cells, which displayed no significant difference in the abundance of each viral mutant at 72 hpi (Fig 8J). These data suggest that in cells that lack both G3BP proteins and cannot produce SGs, high $N^{*M210}$-producing viruses have comparable replication fitness to low $N^{*M210}$-producing viruses. There are two possible interpretations from these data that are not mutually exclusive. (i) $N^{*M210}$ production by $KR^{+TRS}$ confers a fitness advantage, in part, by efficiently antagonizing SGs during infection. Thus, when SGs can no longer form, this fitness advantage is lost. (ii) $N^{*M210}$ antagonizes another antiviral function performed by G3BP1/2, independent of SG formation. Thus, when G3BP1/2 proteins are no longer present, viruses that produce $N^{*M210}$ no longer have the fitness advantage. Because $N^{*M210}$ does not co-precipitate with G3BP1 (Fig 7H), and our data show that the dsRNA-binding ability of $N^{*M210}$ is required for its superior ability to antagonize multiple RNA granules (Figs 7A, 7B, 7I, 7J, S5A, and S5B), we favor the former interpretation.

Taken together, our data show that dominant SARS-CoV-2 variants have evolved to produce the truncated N proteoform ($N^{*M210}$) in greater abundance, an event that enhances viral fitness by antagonizing multiple arms of the antiviral

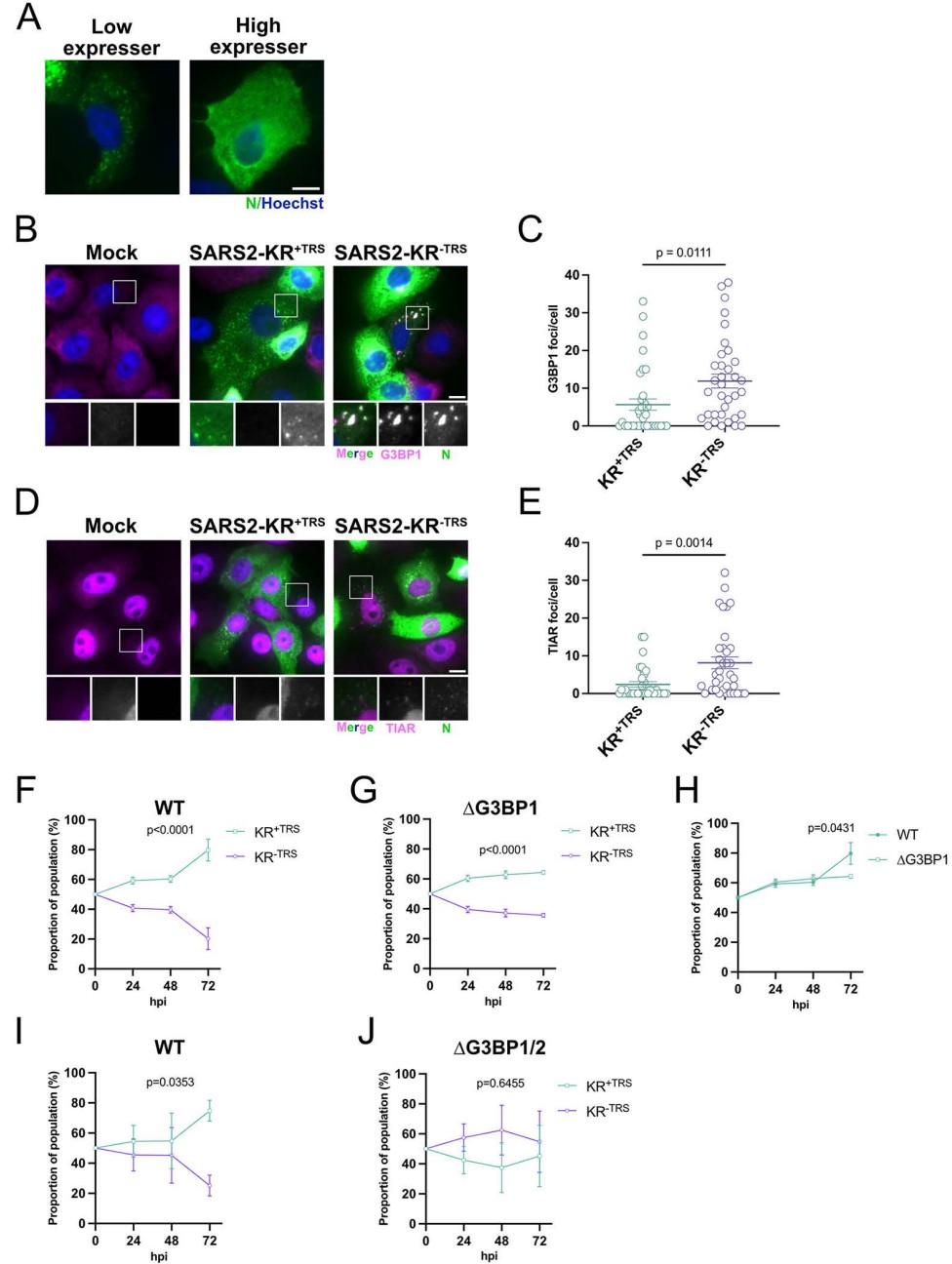

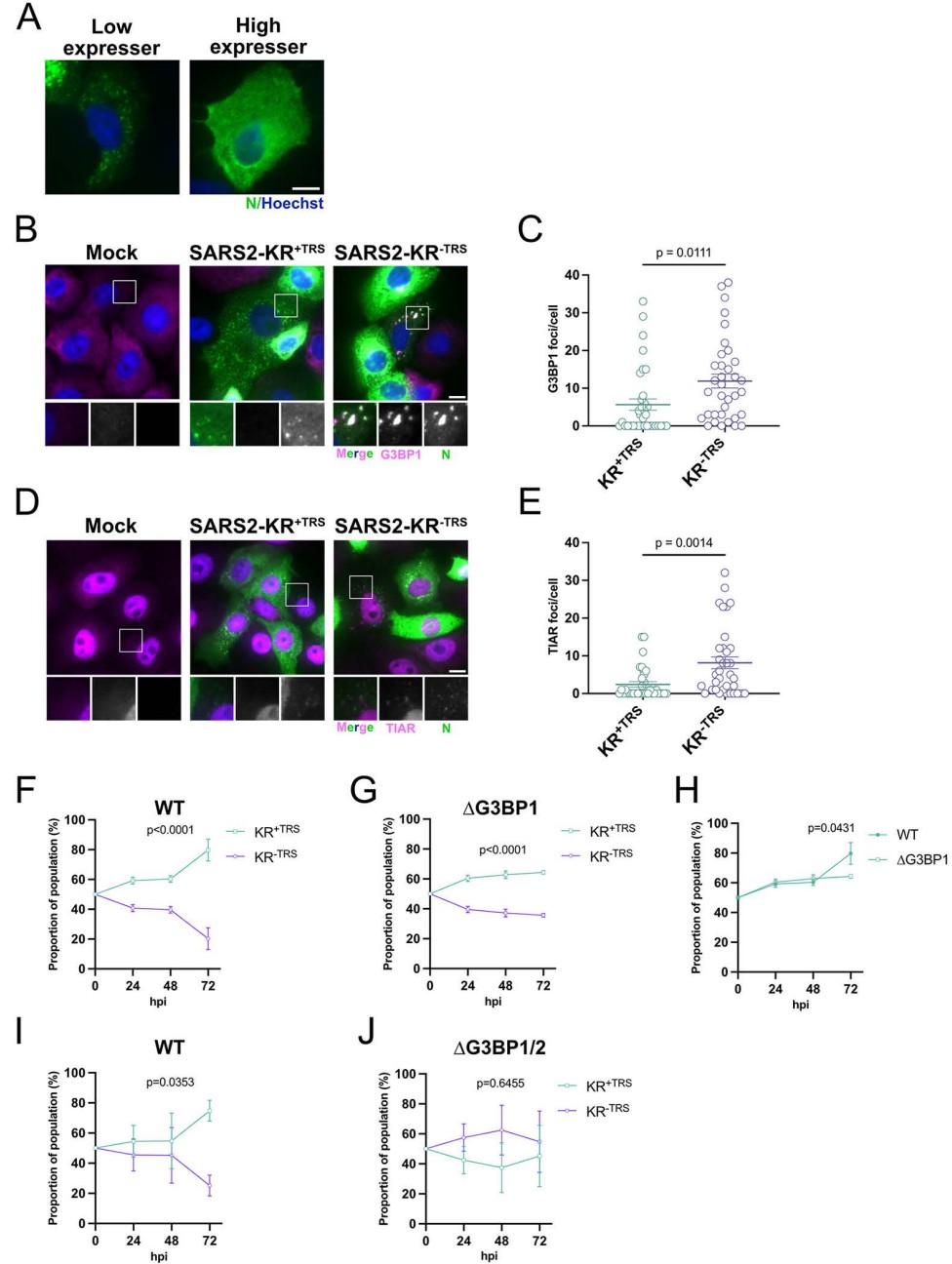

**Fig 8. N\*M210 promotes viral fitness by antagonizing G3BP1 foci. A.** VeroE6 cells were infected with rSARS-CoV-2 KR+TRS (MOI = 1). At 6 hpi, cells were fixed and immunostained with a SARS-CoV-2 N antibody (Alexa 488). Nuclei were stained with Hoechst. Maximum intensity projections are presented here as representative images of the heterogenous infection and named 'low expressors' and 'high expressors' based on N expression. Scale bar = 10 μm. **B–E.** VeroE6 cells were infected with rSARS-CoV-2 KR+TRS or rSARS-CoV-2 KR−TRS (MOI = 1), or mock-infected. At 6 hpi, cells were fixed and immunostained with a SARS-CoV-2 N antibody (Alexa 488) and G3BP1 antibody (B) or TIAR antibody (D) (Alexa 647). Nuclei were stained with Hoechst. Images were captured using a Zeiss AxioObserver Z1 microscope with the 40X oil immersion objective. Scale bar = 10 μm. Images were blinded and G3BP1/TIAR foci were quantified in 'low expressor' N-positive cells manually. Thirty-five cells were measured per condition. Each datapoint represents a single cell. Statistics were performed using an unpaired *T* test; mean; SEM. See S1 Data for quantification values. **F, G.** Wildtype (WT) or G3BP1 knockout (ΔG3BP1) HEK293ACE2 cells were coinfected with equal infectious titers of KR+TRS and KR−TRS recombinant SARS-CoV-2 viruses to achieve a total MOI of 0.01. Time 0 represents the inferred proportions of each recombinant based on infectious titer input. Intracellular RNA was harvested at the indicated times post-infection and subjected to probe-based RT-qPCR to differentiate recombinant virus abundance. These data represent four independent biological replicates (*n* = 4). Statistics were performed using a simple linear regression; SEM. See S1 Data for quantification values.

**H.** The relative proportions of the KR⁺TRS virus in WT and ΔG3BP1 cells (from the coinfection assays from F and G) were overlaid to compare differences in fitness. Differences in slopes were compared using a simple linear regression; SEM. See S1 Data for quantification values. **I, J.** Wildtype (WT) or G3BP1 and G3BP2 knockout (ΔG3BP1/2) A549^ACE2-TMPRSS 2 cells were coinfected with equal infectious titers of KR⁺TRS and KR⁻TRS recombinant SARS-CoV-2 viruses to achieve a total MOI of 0.02. Time 0 represents the inferred proportions of each recombinant based on infectious titer input. Intracellular RNA was harvested at the indicated times post-infection and subjected to probe-based RT-qPCR to differentiate recombinant virus abundance. These data represent four independent biological replicates (n = 4). Statistics were performed using a simple linear regression; SEM. See S1 Data for quantification values.

response. Using direct viral competition assays in cells deficient for SG formation or interferon signaling, our data suggest that the fitness advantage provided by N*^M210 is largely due to its ability to block SGs/G3BP1 foci formation.

## Viruses with the N* TRS have repeatedly dominated in human populations

Given that the N* TRS mutation confers a fitness advantage through the production of N*^M210 (Fig 3), we wondered if the N* TRS has been selected for in viruses circulating in the human population. If the N* TRS confers a fitness advantage in humans, we would expect this mutation to persist in SARS-CoV-2 isolates. Mears and colleagues (2025) recently showed that the N* TRS mutation was acquired through a triple-substitution (G28881A, G28882A, G28883C), where 28882A and 28883C make the ACGAAC core TRS motif (Fig 9A) [35]. They showed that this mutation was present in all Omicron lineages as of mid-2024 [35]. To determine if the N* TRS has persisted in SARS-CoV-2 evolution, a representative dataset was extracted from Nextstrain.org from 2019 to 2025 [72]. Virus genomes were binned into N* TRS-positive or negative by searching for the TRS core sequence within the nucleocapsid gene. The prevalence of viral sequences containing the N* TRS was plotted over time (Fig 9B). Since the N* TRS mutation first emerged in early 2020, there were four major fluctuations in its prevalence between 2020 and 2025: 1. N* TRS-containing viruses were highly prevalent (>50%) in mid 2020 and early 2021 [20]; 2. In late 2021, N* TRS-containing viruses were undetectable in our dataset coinciding with the rise of the Delta variant; 3. Between early 2022 and late 2024, N* TRS-containing viruses were nearly fixed in the population, with ~95 percent of all viral sequences containing the N* TRS; and 4. In early 2025, the prevalence of N* TRS-containing viruses dropped to below 50 percent, then rapidly returned to dominance again. The domination and disappearance of N* TRS-containing viruses suggests that N*^M210 confers a context-dependent fitness advantage. Amongst a multitude of co-occurring mutations in SARS-CoV-2, mutations in other genes may offset the fitness advantage conferred by N*^M210.

The fluctuations in the prevalence of the N* TRS can be explained through two different mutations: G28881A/G28882A/G28883C and G28884C. The G28881A/G28882A/G28883C triple mutation first appeared in early 2020 enabling the formation of the N* TRS [18,35] and caused the R203K/G204R amino acid substitution [20,23]. The G28881A/G28882A/G28883C mutation persisted in clade 20B descendants, including Alpha, Gamma, and the Omicron lineages (Fig 9C). The G28881A/G28882A/G28883C mutation has remained near fixation since the emergence of Omicron in late 2021 (Fig 9C). This means that the decline in N* TRS-containing viruses observed in late 2024/early 2025 must be due to a unique mutation.

In late 2024, the XEC-Omicron variant emerged (Nextstrain.org). XEC contained a G28884C mutation, interrupting the canonical TRS core sequence and causing a G204P amino acid change in N^FL (Fig 9A). Mapping the prevalence of G28884C, we found that the rise of XEC explained the loss of N* TRS-containing viruses, as the prevalence of G28884C-containing sequences was inversely proportional to those containing the N* TRS (Fig 9B and 9D). However, the loss of the N* TRS was short-lived; the G28884C mutation rapidly disappeared and was replaced by N* TRS-containing viruses. As of September 2025, N* TRS-containing viruses compose ~98 percent of circulating SARS-CoV-2 viruses in the human population (Fig 9B). Using data derived from circulating SARS-CoV-2 strains in humans, we show that N* TRS-containing viruses have not remained fixed in the population, but have repeatedly outcompeted non-TRS-containing viruses, supporting a model where N*^M210 confers an evolutionary fitness advantage in humans.

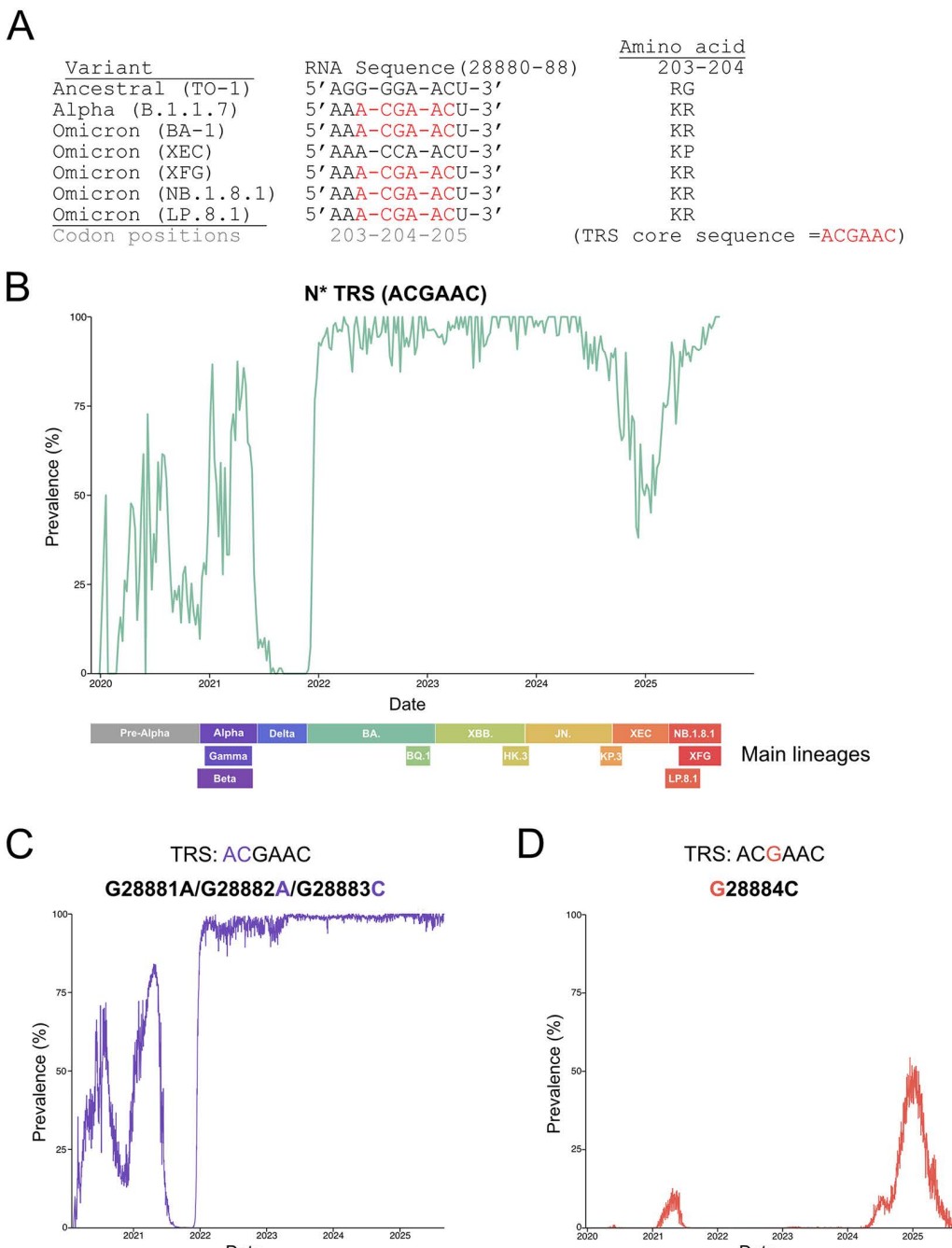

**Fig 9. N\* TRS-containing variants continue to persist in circulating SARS-CoV-2 isolates. A.** Summary of SARS-CoV-2 variant sequence diversity at the mutational hotspot in the N gene where canonical TRS core sequence insertion occurred. Sequence of the TRS hexamer is labeled in red. Codon and nucleotide positions based on the ancestral sequence. **B.** The prevalence of N\* TRS-containing viruses was determined by extracting an aligned SARS-CoV-2 sequence dataset from Nextstrain.org. The dataset was temporally balanced analyzing 2000 representative genomes per year for 2020−2025. Genomes were binned based on the presence of the core sequence of the TRS (ACGAAC) within the N gene. Prevalence was calculated by dividing the total number of N\* TRS-containing genomes by the total number of date-matched genomes. Time was plotted per week. Data analysis and representation were created using RStudio. The approximate dates of relevant SARS-CoV-2 lineages were mapped using Nextstrain.org. **C.** The prevalence of the G28881A/G28882A/G28883C mutation was determined by extracting SARS-CoV-2 sequences from Nextstrain.org. Genomes with the G28881A/G28882A/G28883C mutation were plotted over time. Only days with >30 sequences collected per day were considered to balance the data where sequencing was limited (see S6A Fig). 28882A and 28883C are highlighted in purple as they contributed to the N\* TRS. **D.** The prevalence of the G28884C mutation was determined as in Fig 9C. G28884 is highlighted in orange as it contributed to the N\* TRS.

## Discussion

The host-pathogen relationship during a viral infection can be likened to a molecular game of chess, where each offensive move is rapidly countered until one side ultimately wins. However, rather than pushing pieces on a board, the "moves" are genetic acquisitions, which gives either the virus or the cell the edge to succeed. There are two ways mutations can improve virus fitness; increase the efficiency of the virus replication cycle or decrease the effectiveness of the host antiviral response. Here, we show that a genetic acquisition by SARS-CoV-2 enhances the synthesis of a truncated nucleocapsid proteoform ($N^{*M210}$) to give select variants a competitive edge. We characterize the mechanism of $N^{*M210}$ and show that during infection and overexpression, $N^{*M210}$ antagonizes multiple branches of the cellular antiviral response. Our major findings are as follows: (i) SARS-CoV-2 has acquired mutations to increase the production of $N^{*M210}$; (ii) $N^{*M210}$ is a dsRNA-binding protein that diminishes the interferon response and blocks antiviral granules; (iii) $N^{*M210}$ production confers a fitness advantage in cells and in vivo; (iv) The fitness advantage is lost in $\Delta G3BP1/2$ cells, consistent with the hypothesis that $N^{*M210}$ increases virus fitness, at least in part, by antagonizing SGs.

### SARS-CoV-2 produces truncated N proteoforms

Viruses have genomes that are magnitudes smaller than mammalian genomes, yet despite this, viruses still efficiently hijack and undermine cellular defences. One of the ways viruses circumvent this imbalance is by maximizing their genetic economy. For example, many viruses produce truncated versions of their proteins, which can alter protein function without requiring additional genetic space [42,73–75]. Here, using overexpression and infection, we show that SARS-CoV-2 produces multiple truncated versions of the N protein, including $N^{*M210}$, $N^{*M234}$, and $N^{*37kDa}$. Our results show that internal translation initiation from methionine codons in positions 210 and 234 produces $N^*$ proteoforms $N^{*M210}$, and $N^{*M234}$ (Figs 2K, S1E, and S1F). Although the N gene also contains a putative in-frame start codon at M101, we did not detect this $N^*$ proteoform after overexpression (Fig 2K). This suggests that the M101 AUG is suboptimal for translational initiation, or that the protein product is unstable. We suspect that $N^{*37kDa}$ is a cleavage product that either requires other viral proteins or the induction of an antiviral state to form as it is only detectable by immunoblotting of infected cell lysate and not lysate after ectopic expression of $N^{FL}$.

### $N^{*M210}$ is a viral dsRNA-binding protein that inhibits dsRNA-induced antiviral responses

The N protein contains two structured domains that can bind RNA, the NTD and CTD; the NTD has an affinity for TRS-like ssRNA and the CTD, containing the dsRBM, prefers dsRNAs [33,34]. Because $N^{*M210}$ contains the entire CTD, we expected $N^{*M210}$ to retain dsRNA binding activity. We found that $N^{*M210}$ interacts with multiple types of dsRNA, including small 18-nucleotide duplexes, and has minimal co-localization with ssRNA (Fig 4E). $N^{FL}$ did not readily co-localize with ssRNA; however, it is possible that the 18 bp ssRNA probe is not the optimal length or sequence for $N^{FL}$ binding [34,76]. Using orthogonal approaches, we show that $N^{*M210}$ has an increased ability to bind dsRNA compared to $N^{FL}$ despite sharing an identical dsRBM (Figs 4, S3, and S4). If both $N^{FL}$ and $N^{*M210}$ share the same dsRBM, why would $N^{*M210}$ have increased ability to interact with dsRNA? One important difference between $N^{FL}$ and $N^{*M210}$ is that $N^{*M210}$ does not contain the SR region and therefore is likely not subjected to hyperphosphorylation [32,77]. Hyperphosphorylation of the SR region alters the RNA-binding properties of N [78]. Therefore, it is possible that $N^{*M210}$ acts as a "phospho-dead" version of $N^{FL}$, which allows $N^{*M210}$ to have constitutive dsRNA binding properties.

We found that $N^{*M210}$ represses antiviral pathways by sequestering dsRNA. Although overexpressed $N^{*M210}$ has an increased ability to interact with dsRNA compared to $N^{FL}$, $N^{*M210}$ was not significantly better at blocking the activation of PKR and IRF3/IFNβ (Fig 6). This is likely a result of the excess amount of $N^{FL}/N^{*M210}$ made during overexpression compared to dsRNA, which could mask the potential differences between these two proteoforms. It is also possible that the weaker dsRNA binding ability of $N^{FL}$ is sufficient to inhibit poly I:C-induced antiviral responses. However, during an

infection, N*$^{M210}$ plays a significant role in antiviral antagonism as evidenced by the finding that KR$^{+TRS}$ (high N*$^{M210}$ production) is superior at inhibiting interferon induction compared to KR$^{-TRS}$ (Fig 6N). N$^{FL}$ is required for viral transcription and genome packaging, while it moonlights as an antiviral antagonist. By contrast, N*$^{M210}$ lacks the N-terminal 209 amino acids of N$^{FL}$, liberating it from these other functions to act as a dedicated dsRNA-binding protein and antiviral response antagonist.

## Other possible functions of N*$^{M210}$

In this study, we focus on the functions of N*$^{M210}$ as an immune antagonist; however, N*$^{M210}$ may also participate in other aspects of the viral infectious cycle. Recently, the Morgan and Doudna labs found that N*$^{M210}$ participates in viral packaging [77,79]. In support of this finding, we show that N*$^{M210}$ and other N* proteoforms are found in extracellular viral particles (Fig 2J). To our knowledge, we are the first group to show that truncated N proteoforms are packaged into authentic SARS-CoV-2 virions, raising the interesting possibility that packaged N*$^{M210}$ could play a role during the early phase of viral infection. Furthermore, Bouhaddou and colleagues (2023) found that N*$^{M210}$ interacts with the RNA polymerase II-associated factor (PAF) complex, although the biological consequences of N*$^{M210}$-PAF interaction are unknown [80]. These data suggest that N*$^{M210}$ should localize to the nucleus. In agreement with this, we observe that a subset of N*$^{M210}$ adopts nuclear localization after ectopic expression (Fig 4C); however, N*$^{M210}$ nuclear localization during authentic infection remains unconfirmed.

## N*$^{M210}$ antagonizes RNP granules via its dsRBM

RNP granules, including SGs, RLBs, and P-bodies, are emerging as components of a robust antiviral response. Many viruses, including SARS-CoV-2, transiently induce SGs during the early stages of infection, until viral proteins potently suppress their formation [15,81–83]. Although most viruses rapidly suppress SG formation, paradoxically, there are a few examples where SG formation promotes virus replication [84,85]. There are three common approaches viruses use to block SG formation: (i) preventing eIF2α phosphorylation [86–88]; (ii) sequestering or destroying G3BP1/2 [16,82,89]; (iii) degrading host mRNA [11,90]. Recently, the Parker lab has shown that RNA-RNA interactions are critical for SG formation and stability [91,92]. Proteins that promote these RNA-RNA interactions, like G3BP1, are called 'condensers', whereas proteins that inhibit RNA-RNA interactions, such as ADAR1 and eIF4A, are called 'decondensers' [93–95].

SARS-CoV-2 N$^{FL}$ inhibits SGs by preventing the activation of PKR, sequestering dsRNA, and sequestering G3BP1/2 [31,43,60–62]. G3BP1 foci form in response to SARS-CoV-2 infection and it is thought that they restrict CoV replication [15]. For example, SG inhibition via G3BP1 knockdown/knockout increased SARS-CoV-2 replication in vitro and in vivo [31,59]. Similarly, Dolliver and colleagues (2022) found that overexpression of G3BP1 increased SG formation and correspondingly decreased the replication of the common cold coronavirus HCoV-OC43 [11]. Precisely how SGs exercise their antiviral activity remains unclear. Uncoupling the proposed antiviral functions of G3BP1 condensates versus the G3BP1 protein is an ongoing challenge in the field. Burke and colleagues (2024) found that G3BP1 can trap and repress SARS-CoV-2 RNAs [8]. In a complementary study, Yang and colleagues (2024) illustrated that loss of the G3BP1-interacting motif in N diminished SARS-CoV-2 replication, supporting a model where the N-G3BP1 interaction is necessary for optimal replication by preventing viral RNA translational repression [62].

Consistent with the literature, we show that overexpressed N$^{FL}$ blocks sodium arsenite-induced SGs; yet we show that optimal SG inhibition in this assay requires the dsRBM (Fig 7I and 7J). Because RNA binding increases the interaction between G3BP1 and N$^{FL}$ [59], it is possible that in addition to the G3BP1-interacting motif, N$^{FL}$ requires the dsRBM to block SGs. Despite lacking the G3BP1-interacting motif, N*$^{M210}$ overexpression also blocked SA-induced SGs, more potently than N$^{FL}$ (Fig 7I and 7J). N*$^{M210}$-mediated SG inhibition was completely dependent on the dsRBM. These data indicate that protein interaction with G3BP1 is not always an essential feature of SG inhibition. Furthermore, because SA-induced SGs form in the absence of exogenous dsRNA, these data are consistent with a model wherein N*$^{M210}$ inhibits

SGs by interacting with cellular RNAs, preventing RNA-RNA interactions necessary for granule formation and stability. If N*M210 inhibits RNA condensation, it should be granule-agnostic. In support of this theory, we show that N*M210 also inhibits P-bodies in a dsRBM-dependent mechanism (S5A Fig).

The relationship between NFL and RLBs is less well understood, partly because RLBs have only recently been discovered and these granules were often misidentified as SGs in the past (both contain G3BP1). Although RLBs often contain G3BPs, unlike arsenite-induced SGs, G3BPs are not essential for their formation [58]. Instead, RLB formation requires the activation of RNase L and presumably the condensation of cleaved RNAs with RBPs. Here, we find that NFL and N*M210 dampen RNase L activation (Fig 6I); however, only N*M210, not NFL, significantly reduced RLB formation in HeLa cells (Fig 7B). One explanation for this discrepancy may be that poly I:C is a weak RLB inducer in HeLa cells, and most cells remain RLB-negative, irrespective of N expression level (Fig 7B). It is also possible that N*M210 blocks RLB formation by preventing the condensation of cleaved RNAs with RBPs, a mechanism like that proposed for SG and P-body dissolution.

In addition to N, SARS-CoV-2 encodes other proteins that have been shown to inhibit G3BP1 foci formation, including Nsp1 [11] and Nsp15 [87]. Given the arsenal of G3BP1 foci inhibitors encoded by this virus, it is not surprising that G3BP1 foci are only observed during early stages of infection [15]. The initial stages of an infection are pivotal and often dictate the outcome of the infection [96]. If incoming genomic RNA is sequestered in a SG, it will not be translated, and this would result in a failed infection. We show that in infected cells when N is limiting, KR+TRS (high N*M210) blocks G3BP1 foci formation better than KR−TRS (low N*M210) (Fig 8B–8E). We speculate that as infection proceeds, both KR+TRS and KR−TRS viruses will produce other viral proteins that can inhibit G3BP1 foci formation, masking the impact of N*M210. This begs the question, why is a greater amount of N*M210 advantageous if the virus encodes other granule inhibitors? Given that N*M210 can be packaged into virions (Fig 2J), we propose that N*M210 diminishes granule formation immediately post-entry, preventing viral genomic RNA sequestration in G3BP1 foci. As the superior antagonist of these foci, N*M210 is specifically poised for this task, and because of this, viruses that produce N*M210 in greater abundance initiate more successful infections, resulting in greater replication fitness.

## N*M210 production increases virus replication fitness

All SARS-CoV-2 variants tested produced multiple truncated N proteoforms; however, N*M210 and N*M234 production is upregulated in Alpha, Gamma, and Omicron due to the internal TRS acquisition. This suggests that N* production may be undergoing positive selection due to its importance for SARS-CoV-2 replication in humans (Fig 9). To directly determine if N*M210 production impacts virus fitness, we conducted co-infection competition assays and found that the addition of a TRS internal to the N gene, which promotes N*M210 synthesis, confers a replication advantage in multiple cell lines, primary human cells, and in K18ACE2 mice (Figs 3, 6, 8, and S2). Previous studies examined the consequence of the R203K/G204R amino acid substitution in NFL and showed that this mutation conferred a fitness advantage by enhancing NFL phosphorylation, phase separation, and antiviral suppression [20,23,97]. However, the R203K/G204R amino acid change also introduces the internal TRS, which upregulates N*M210 synthesis, and these studies did not consider the impact of increased N*M210 production on viral fitness. We directly answer this question using recombinant viruses to uncouple the role of the new internal TRS and the role of altered NFL activity due to amino acid changes. We reveal that the TRS, and corresponding upregulation of N*M210 production, increases virus fitness independent of amino acid changes, as rSARS-2 KR+TRS outcompeted both the WT and the amino acid-matched KR−TRS viruses. We did not observe a fitness difference between KR−TRS and WT, suggesting that in our system, the R203K/G204R amino acid substitution did not influence virus fitness (Fig 3L). Furthermore, we did not observe a fitness difference between the WT and the M210I viruses, suggesting that basal production of N*M210 was insufficient to improve virus replication fitness in vitro. It is possible that in the context of a viral infection in an animal model, low levels of N*M210 produced by internal translation initiation may influence viral fitness.

We consider three different mechanisms by which the N*$^{M210}$ protein could improve viral fitness. (i) Because N*$^{M210}$ inhibits the activation of RNase L and PKR, N* could prevent translation arrest (Fig 6). (ii) N*$^{M210}$ could promote viral fitness by limiting the interferon response. Consistent with this, infection with KR$^{+TRS}$ (more N*$^{M210}$) blocked IFNβ induction more than KR$^{−TRS}$ (less N*$^{M210}$) (Fig 6N). However, KR$^{+TRS}$ still outcompeted KR$^{−TRS}$ in VeroE6 cells, which are deficient in the interferon response, making this unlikely to be the only antiviral activity of N*$^{M210}$. (iii) Because N*$^{M210}$ can inhibit SG formation triggered by sodium arsenite, N*$^{M210}$ could promote viral fitness by blocking RNP granule formation (Fig 7). Consistent with this, N*$^{M210}$ provides a fitness advantage, in part, by antagonizing G3BP1 foci, as the fitness difference between KR$^{+TRS}$ and KR$^{−TRS}$ in WT cells was diminished in ΔG3BP1 cells and lost in ΔG3BP1/2 cells (Fig 8). Together, these data indicate that although N*$^{M210}$ may promote virus fitness in multiple ways, one key mechanism is via antagonizing SG or RLB formation. Because G3BPs are found in RLBs but are not required for RLB formation, there are three possible explanations for the fitness loss in G3BP knockout cells: (i) The G3BP1 protein, not G3BP1 foci, exerts an antiviral role that is antagonized by N*$^{M210}$; (ii) G3BP knockout eliminates SGs, rendering N*$^{M210}$ superfluous; or (iii) G3BP knockout does not block RLB formation, but by preventing G3BP localization to RLBs, antiviral capabilities of RLBs are limited.

During the writing of this manuscript, Mears and colleagues (2025) published a complementary study using a similar recombinant virus strategy, supporting the finding that the acquisition of the novel TRS in N improves virus fitness [35]. They also showed that N*$^{M210}$, which they call N.iORF3, impacts fitness, in part, by inhibiting RIG-I activation. However, in this study, the mechanism of RIG-I inhibition was not determined [37]. Our data support and extend their observations by providing a mechanistic explanation for how N*$^{M210}$/N.iORF3 antagonizes RIG-I activation, namely by sequestering dsRNA. Our work also shows that N*$^{M210}$ antagonizes multiple antiviral pathways, including not only RIG-I/IFN but also PKR, RNase L, P-bodies, RLBs, and SGs to promote SARS-CoV-2 replication. We also expand on the observations of Mears and colleagues (2025) to show that N*$^{M210}$ provides a fitness advantage in vivo, data that support the hypothesis that N*$^{M210}$ enhances replication in humans [35]. Consistent with this hypothesis, we show that N* TRS-containing viruses have undergone large fluctuations yet ultimately continue to persist and circulate in the human population at the time of this writing, suggesting N* TRS-containing viruses have a fitness advantage in the human population (Fig 9). There are two important caveats to discuss. First, from population data, it is impossible to uncouple if the N* TRS or the amino acid substitution in N$^{FL}$ (R203K/G204R) is the primary driver of fitness in the human population. However, given that the N* TRS mutation consistently enhanced fitness (Figs 3, S2, 6, and 8), while the R203K/G204R amino acid substitution did not (Fig 3L), we propose that the TRS acquisition is likely the primary driver of virus fitness. Second, because each of these variants have a constellation of mutations, it is difficult to uncouple the extent to which the TRS acquisition/ N*$^{M210}$ production contributed to the evolutionary takeover of circulating strains. In other words, is N*$^{M210}$ a cause of variant dominance, or merely a correlation? Given the transient disappearance of the N* TRS in late 2024, and its reappearance and dominance in the human population in 2025 (Fig 9), we speculate that although this mutation confers a fitness advantage, it can be influenced and at times overshadowed by other co-occurring mutations.

Here, we characterize a mutation that arose in the SARS-CoV-2 nucleocapsid gene. We show that this mutation enables the synthesis of a new viral sgRNA, increasing the production of a truncated N protein, N*$^{M210}$, a potent dsRNA-binding protein capable of antagonizing the cellular antiviral response. Using rSARS-2, we show that the upregulation of N*$^{M210}$ increases viral antagonism of the host response and promotes viral fitness in vitro and in vivo, giving these viruses an edge to outcompete other variants.

## Limitations of this study

We acknowledge the following limitations of our study. (i) Because the anti-N antibody epitope is in the CTD, we were only able to detect amino-terminally truncated proteoforms. Therefore, all potential carboxy-terminally truncated products were not detected. (ii) We conducted competition assays in ΔG3BP1 and ΔG3BP1/2 cells in different cell types, and although these yielded similar results, supporting our conclusions, the knockout of any gene can result in off-target effects

or compensatory mutations that may also influence virus fitness. Future studies should complement knockout cell lines with G3BP1/2 to confirm these findings. (iii) Co-infection competition assays measure the proportion of a viral species over time. The dominance of a certain species may reflect increased antiviral tolerance/evasion or increased replication kinetics or efficiency. In short, if a viral species can replicate faster, it will likely outcompete its slower counterpart in these co-infection assays. A second limitation of the co-infection competition assay is that there is potential for cooperative use of N*M210 by both recombinants if they co-infect the same cell. In such cases, N*M210 produced by KR+TRS could be used to enhance replication of both viral populations; for this reason, these assays are likely to underestimate the fitness advantage of KR+TRS, further highlighting the significance of N*M210 in virus replication.

## Materials and methods

### Animal use ethics statement

Animal work was conducted under CL3 conditions with authorization from the CRCHU de Quebec-Université Laval animal care committee (#2025-1709) were approved by the Comité de Protection des Animaux de l'Université Laval (CPAUL-3).

### Cell culture

All cells were grown in a humidified chamber at 37 °C, 5% $CO_2$, and 20% $O_2$. A549 (ATCC), A549-ΔG3BP1/2 (a generous gift from Dr. James Burke, University of Florida Scripps Institute), HEK293T (ATCC), HEK293A (ATCC), HEK293A-ΔG3BP1 (a generous gift from Dr. Denys Khaperskyy, Dalhousie University), HeLa (ATCC), VeroE6 (ATCC), and VeroE6TMPRSS2 cells (a generous gift from Dr. Michael Joyce and Holly Bandi, University of Alberta) were cultured in DMEM (Thermo-Fisher) supplemented with 10% heat-inactivated fetal bovine serum (Thermo-Fisher) and penicillin (100 U/mL), streptomycin (100 U/mL), L-glutamine (2 mM) (PSQ; Thermo-Fisher). Calu3 (ATCC) cells were cultured in EMEM (ATCC) supplemented with 10% heat-inactivated fetal bovine serum (Thermo-Fisher) and penicillin (100 U/mL), streptomycin (100 U/mL), L-glutamine (2 mM). HUVECs (Lonza) were cultured in EGM2 (Lonza). HUVECs were seeded onto glass coverslips or plastic dishes coated in 0.1% (w/v) porcine gelatin (Sigma) in 1× PBS (Thermo-Fisher). HUVECs were used between passages 5–7.

A549ACE2, A549ACE2-TMPRSS2, A549-ΔG3BP1/2ACE2-TMPRSS2, HEK293AACE2, and HEK293A-ΔG3BP1ACE2 cells were generated by lentivirus transduction of ACE2 or ACE2-TMPRSS2. Cells were selected for 48–72 hours in 5 µg/mL blasticidin or 1 µg/mL puromycin (Thermo-Fisher). Following selection, cells were cultured in DMEM with 10% FBS and 1× PSQ. HUVECACE2 cells were generated by lentivirus transduction of ACE2 at passage 5, selected with 5 µg/mL blasticidin for 48 h, then used for virus infections at passage 6.

### Plasmids and cloning

SARS-CoV-2 codon-optimized, FLAG-tagged nucleocapsid (pLJM1-NFL-FLAG; [14]) was subcloned into pcDNA3.1 and pLVX-IRES-puro backbones. N truncations (N*) were generated by PCR amplification and restriction digestions using BamHI and EcoRI (NEB) restriction sites. N amino acid substitutions were generated by site-directed mutagenesis of the pcDNA3.1-NFL-FLAG or pcDNA3.1-N*M210-FLAG plasmids. Methionine substitutions were selected based on conserved residues in related beta-CoV HCoV-OC43. pcDNA3.1-NFL-HA and pcDNA3.1-N*M210-HA were cloned from pcDNA3.1-NFL-FLAG and pcDNA3.1-N*M210-FLAG, respectively. To generate standards for qPCR, a 153 bp fragment of ORF1ab was cloned from SARS-CoV-2-infected HUVECACE2 cells and subcloned into pcDNA3.1 using BamHI and EcoRI. Standards for co-infection competition assays were generated by subcloning fragments from the BACmid (see Materials and methods "recombinant virus cloning and mutagenesis") containing the N open reading frame into pcDNA3.1 using NheI and NotI. All PCRs were performed with Phusion High-Fidelity PCR Master Mix with GC Buffer (New England Biolabs).

## Recombinant virus cloning and mutagenesis

For viral reverse genetics and recombinant virus rescue, we used the infectious cDNA clone bacterial artificial chromosome (BAC) system (a kind gift from Dr. Luis Martinez-Sobrido, Texas Biomedical Research Institute) [37,38]. For N gene mutagenesis, a site-directed mutagenesis overlap extension PCR and Gibson assembly approach was used. For Gibson assembly, the BAC was divided into 4 fragments of 5−6 kb and one larger fragment of 15.4 kb containing the vector backbone. All 5−6 kb fragments were amplified by conventional PCR from the parental BAC containing the SARS-CoV-2 USA-WA1/2020 strain genome described in Ye and colleagues (2020) [38]; the 15.4 kb fragment was generated by restriction digestion. For mutagenesis, the N gene-containing fragment was further broken up into three subfragments; amino acid and TRS mutations were introduced with overlapping forward and reverse mismatch primers and invariant end primers (Table 1), and the full-length mutant fragments were then assembled and amplified by overlap extension PCR. KR$^{+TRS}$ mutation was modeled after the SARS-CoV-2 Gamma variant.

All Gibson assembly fragments were gel-extracted and assembled using the Gibson Assembly Master Mix (New England Biolabs). Fragments were combined at equimolar ratios (15 fmol each) in 30 µL reactions that were incubated at 50 °C for 60 min according to the manufacturer's instructions. All mutant BACs were validated by restriction digest and Sanger sequencing of the entire viral genome using 56 sequencing primers (Table 2). Verified BAC clone DNA was purified for transfection using plasmid Midi Kit (Qiagen). PCR cycling conditions for amplification of the Gibson fragments from the parental BAC: *95 °C - 1 min; 30 × (95 °C - 15 s; 60 °C - 30 s; 72 °C - 5 min); 72 °C - 5 min.* PCR cycling conditions for mutagenesis and sequential overlap extension reactions: *95 °C - 1 min; 15–25 × (95 °C - 15 s; 60–63 °C - 30 s; 72 °C - 2–5 min); 72 °C - 5 min.*

## Recombinant virus generation and validation

Recombinant SARS-CoV-2 were made after consultation with the Public Health Agency of Canada (PHAC), the University of Calgary's Biosafety Office, and the Containment Level 3 (CL3) Oversight Committee. VeroE6$^{TMPRSS2}$ cells were seeded

**Table 1. Amplification and mutagenesis primers (mismatches in uppercase).**

| Primer Name | Sequence (5′ → 3′) |
| --- | --- |
| FragN.1_Rv | cactgctactggaatggtctgtgtttaatttatagttgccaatcctg |
| FragN.2_Fw | caggattggcaactataaattaaacacagaccattccagtagcagtg |
| FragN_cloning_Fw | gctttgctgtatgaccagttgctgtagttgtctc |
| FragN_cloning_Rv | gaatggcagaaattcgatgataagctgtcaaacatgag |
| Gibson_F1_Fw | ttaataattggttgaagcagttaattaaagttacacttgtgttcc |
| Gibson_F1_Rv | agttaccattgagatcttgattatctaatgtcagtacaccaac |
| Gibson_F2_Fw | gataatcaagatctcaatggtaactggtatgatttcggtg |
| Gibson_F2_Rv | cgatatgttcgaaggcatagccttctaatttataccgttc |
| Gibson_F3_Fw | ggctatgccttcgaacatatcgtttatggagattttagtc |
| Gibson_F3_Rv | gtaatgtaatttgactcctttgagcactggctcag |
| Gibson_F4_Fw | gctcaaaggagtcaaattacattacacataaacgaacttatg |
| Gibson_F4_Rv | gctgtcaaacatgagaattggtcgacgg |
| KR$^{-TRS}$_Fw | caactccaggcagcagtaAgAGGacttctcctgctagaatg |
| KR$^{-TRS}$_Rv | ctagcaggagaagtCCTcTtactgctgcctggagttgaatttc |
| KR$^{+TRS}$_Fw | caactccaggcagcTCtaAACgaacttctcctgctagaatg |
| KR$^{+TRS}$_Rv | ctagcaggagaagttcGTTtaGAgctgcctggagttgaatttc |
| N_M210I_Fw | ctcctgctagaatTgctggcaatggcg |
| N_M210I_Rv | gccattgccagcAattctagcaggagaag |

**Table 2. rSARS2-BACmid sequencing primers.**

| Primer Name | Sequence (5′ → 3′) |
|---|---|
| Seq_gF1_cov-1V | attaaaggtttataccttcccagg |
| Seq_gF1_cov-655V | agctggtggccatagttac |
| Seq_gF1_cov-1321V | aggtgccactacttgtgg |
| Seq_gF1_cov-1925V | ctgctcaaaattctgtgcg |
| Seq_gF1_cov-2572V | ctactagtgaagctgttgaagc |
| Seq_gF1_cov-3225V | ctgttggtcaacaagacgg |
| Seq_gF1_cov-528R | agctcaaccataacatgacc |
| Seq_gF2_cov-3824V | gtttcaagctttttggaaatg |
| Seq_gF2_cov-4431V | tgcctgtctgtgtggaaac |
| Seq_gF2_cov-4990V | caacattaacctccacacgc |
| Seq_gF2_cov-555V | acttgtggacaacagcag |
| Seq_gF2_cov-6109V | gaaacctgcttcaagagag |
| Seq_gF2_cov-6737V | acacggtgtttaaaccgtg |
| Seq_gF2_cov-7382V | caaatggcccccgatttcag |
| Seq_gF3_cov-7930V | tcagcgtctgtttactacag |
| Seq_gF3_cov-8481V | cttttaagttgacatgtgcaac |
| Seq_gF3_cov-8995V | atcagcttgtgttttggc |
| Seq_gF3_cov-9534V | ctgtactctgtttaacacc |
| Seq_gF3_cov-10094V | gagggttgtatggtacaag |
| Seq_gF3_cov-10680V | acgctgctgttataaatgg |
| Seq_gF3_cov-11188V | accttctcttgccactg |
| Seq_gF3_cov-11707V | agtttctacacaggagtttag |
| Seq_gF4_cov-12205V | gaagaagtctttgaatgtgg |
| Seq_gF4_cov-12806V | gtacttgcactgttatccg |
| Seq_gF4_cov-13441V | gtcagctgatgcacaatcg |
| Seq_gF4_cov-14062V | gataatcaagatctcaatgg |
| Seq_gF4_cov-14618V | ctacgtgcttttcagtag |
| Seq_gF5_cov-15170V | atcaatagccgccactag |
| Seq_gF5_cov-15677V | acgcatatttgcgtaaac |
| Seq_gF5_cov-16273V | tcattaagatgtggtgcttg |
| Seq_gF5_cov-16853V | gtgatgctgttgtttaccg |
| Seq_gF5_cov-17444V | ctcaattacctgcaccac |
| Seq_F5_cov-18037V | aagctgaaaatgtaacagg |
| Seq_gF6_cov-18588V | tgtcttatgggcacatgg |
| Seq_gF6_cov-19211V | gatatcctgctaattccattg |
| Seq_gF6_cov-19840V | atttgggtgtggacattg |
| Seq_gF6_cov-20459V | aacagatgcgcaaacagg |
| Seq_gF6_cov-20934V | tacgctgcttgtcgattc |
| Seq_gF6_cov-21521V | tgttatttctagtgatgttcttg |
| Seq_gF7_cov-22092V | tggaccttgaaggaaaac |
| Seq_gF7_cov-22685V | tccacttttaagtgttatggag |
| Seq_gF7_cov-23203V | aggcacaggtgttcttac |
| Seq_gF7_cov-23840V | gtacacaattaaaccgtgc |
| Seq_gF7_cov-24428V | cacaagctttaaacacgc |
| Seq_gF8_cov-25068V | tctctggcattaatgcttc |
| Seq_gF8_cov-25624V | cactttgtttgcaacttgc |

*(Continued)*

**Table 2.** (Continued)

| Primer Name | Sequence (5′ → 3′) |
|---|---|
| Seq_gF8_cov-26245V | cattcgtttcggaagagac |
| Seq_gF8_cov-26778V | gtcttgtaggcttgatgtg |
| Seq_gF8_cov-27372V | atggagattgattaaacgaac |
| Seq_gF8_cov-27875V | ttgtcacgcctaaacgaac |
| Seq_gF9_cov-28404V | gtttacccaataatactgcg |
| Seq_gF9_cov-28994V | caacaaggccaaactgtc |
| Seq_gF9_cov-29611V | gtgcagaatgaattctcg |
| Seq_gF9_F7-AvrII-R | gaagtccagcttctggcc |
| N_seq_rCoV2_Fw | caccttttacaattaattgccagg |
| N_seq_rCoV2_Rv | ttctgcacaagagtagactatatcg |

into 6-well plate. Twenty-four hours after seeding, 0.5 µg of recombinant SARS-CoV-2 BAC was combined with 1.5 µL of Lipofectamine 2000 (Thermo-Fisher). The transfection mixture was added to cells containing antibiotic-free DMEM. Once cytopathic effect was observed (48–96 hpi), virus-containing supernatant was harvested, cell debris was removed by centrifugation, and virus was frozen at −80 °C for single use. Viral RNA was harvested from passage 0 and passage 1 stocks using QIAamp Viral RNA Mini kit (Qiagen) as per manufacturer's instructions, without including carrier RNA. Recombinant viruses were propagated in VeroE6$^{TMPRSS2}$ cells. All recombinant virus experiments were conducted with passage 1 virus stocks.

### Next generation sequencing of recombinant viruses

Viral RNA was prepared for whole genome sequencing via multiplexed PCR amplicon tiling using the SARS-CoV-2 Midnight Amplicon Panel [98] (IDT; this study used Midnight Amplicon Panel v1). A detailed protocol is available and was followed up to and including step 11 [99]. Briefly, cDNA was prepared using LunaScript RT Supermix Kit (NEB) according to the manufacturer's specifications, with 8 µL of viral RNA as template. cDNA was then used in two PCR reactions with the Midnight Amplicon Panel to generate two unique amplicon pools which were then mixed at a 1:1 ratio. Combined amplicon pools were submitted to the University of Calgary Centre for Health Genomics and Informatics for library preparation followed by sequencing via Illumina MiSeq 300-cycle nano (2 × 150 reads). Resulting raw single-end sequence Illumina reads were pre-processed using fastp [100] under default parameters for the trimming of Illumina adaptors and low-quality reads; trimmed reads were exported as.fastq.gz files. Trimmed reads were then imported to Geneious Prime (v.2025.0.1) to generate paired end (inward pointing) reads with insert size = "150". Paired reads were further processed using two Genious Prime-resident processing tools: Dedupe (v.38.84 by Brian Bushnell) for the removal of duplicate reads, followed by BBNorm (v.38.84 by Brian Bushnell) [101] for paired read error correction and normalization. Dedupe was run under default parameters, and BBNorm was run under default parameters for error correction and under the following parameters for normalization: Target Coverage Level = "220", Minimum Depth = "6". Fully processed paired reads were then mapped against the SARS-CoV-2 reference genome (NC_045512) which was modified to replace the nucleotides in positions 28,877–28,885 with nucleotide wild cards ("N"). This permits an unbiased sequence alignment considering the mutational differences in this wild card region between WT and mutant viral sequences. Reference mapping parameters: Sensitivity = "Medium Sensitivity/Fast", Minimum mapping quality = "30". The Find Variants/SNPs tool was employed with Minimum Variant Frequency = "0" for detection of all possible sub-populations under the reference sequence wild cards. The wild card region had a coverage of ~330 across sequenced viral species. Consensus sequences were exported with a consensus threshold set to 75%. The consensus sequence for WT virus matched the reference genome with ≥99.4%

likeness across the wild card region; the consensus sequence for mutant viruses matched the expected sequences based on the intended mutations made with mutational variant frequencies ≥97.2% across the wild card region, for all recombinant viruses. Recombinant virus genome sequences were aligned, and sequence ends were trimmed due to poor sequence quality or repetitive sequences. No off-target protein-coding mutations were detected. Genomes were submitted to NCBI: accession numbers: WT: PV935594.1, KR$^{+TRS}$: PV952734, KR$^{-TRS}$: PV936453, M210I: PV955084.

## Mouse infection and sample processing

Animal work was conducted under CL3 conditions at the Université Laval. Viral stocks (rSARS-CoV-2 KR$^{+TRS}$ and KR$^{-TRS}$) obtained from the Calgary team were propagated on Vero cells and titrated as previously described [102]. For infection, groups of 11-week-old 4–5 adult male and female K18$^{ACE2}$ mice were anesthetized with isoflurane and infected by intranasal delivery of 25 µl of saline solution containing 2,500 TCID$_{50}$ of rSARS-CoV-2 KR$^{+TRS}$ and 2,500 TCID$_{50}$ of rSARS-CoV-2 KR$^{+TRS}$ viruses. On day 1 and 3 post-infection, mice were euthanized, and the lung's left inferior lobe was used for total RNA extraction using the Bead Mill Tissue RNA Purification Kit and the Omni Bead Ruptor Bead Mill homogenizer (Kennesaw, GA), as previously described [103]. RNA samples were kept frozen at −80 °C until assayed.

## TaqMan RT-qPCR analysis of viral co-infection competition assays

Determination of transcript viral copy number in viral co-infection competition assays was conducted via a probe-based RT-qPCR TaqMan assay to differentiate viral species. Primer and TaqMan probe sequences can be found in Table 3; all probes contain a 5′ fluorescent reporter and a 3′ non-fluorescent quencher (NFQ) and minor groove binding (MGB) protein. Qubit 1× dsDNA High Sensitivity Assay Kit (Invitrogen) was used to accurately determine competition assay standard plasmid concentrations with a Qubit 3.0 Fluorometer (Invitrogen). The plasmid copy number of standards was calculated and used to prepare a 6-point standard curve between 1E2 and 1E7 known plasmid copies under a 1:10 dilution series. TaqMan probes were validated against their respective standard curve target; reaction efficiencies for all probes were between 90% and 105% with $R^2$ values >0.9990. Probes were also confirmed to be specific for their respective standard target, displaying no cross-reactivity when tested against off-target standards.

For in vitro (cell culture) co-infection competition assay samples, singleplex RT-qPCR reactions (single probe per reaction) were prepared in quadruplicate per samplethen divided into a duplicate for detection of one viral species and a duplicate for detection of the other viral species. Reactions consisted of (final concentrations) 500 nM forward and reverse SARS-CoV-2 N TaqMan assay primers, 250 nM of respective probe, 5 µL 2× TaqMan Universal PCR Master Mix (Thermo-Fisher) containing uracil-N-glycosylase (UNG) for the prevention of carryover contamination, and either 2.5 µL of plasmid standards at 0.4E2–0.4E7 plasmid copies/µL (final of 1E2–1E7 plasmid copies in reaction) or 2.5 µL of diluted cDNA sample. In general, cDNA was diluted 1:50 for HUVEC- originating samples, 1:100 for A549-originating samples, and 1:500 for HEK293A-ΔG3BP1- and VeroE6-originating samples. Reaction cycling conditions were as follows: *UNG activation at 50 °C for 2 min, initial denaturation at 95 °C for 10 min, cycling at 40× (denaturation at 95 °C - 15 s, annealing*

**Table 3. TaqMan primer sequences.**

| Target | 5′ Modification | Probe Sequence (5′ → 3′) | 3′ Modification |
|---|---|---|---|
| N WT | 6-FAM | CAGCAGTAGGGGAAC | MGB-NFQ |
| N KR$^{-TRS}$ | 6-FAM | CAGCAGTAAGAGGAC | MGB-NFQ |
| N KR$^{+TRS}$ | VIC | CAGCTCTAAACGAAC | MGB-NFQ |
| N M210 | 6-FAM | TGCTAGAATGGCTGGC | MGB-NFQ |
| Mut N M210I | VIC | TGCTAGAATTGCTGGC | MGB-NFQ |

*and amplification at 60 °C - 30 s).* For in vivo (mice) co-infection competition assay samples, duplex RT-qPCR reactions (two probes per reaction) were prepared in duplicate for the concurrent detection of both viral species. Reactions consisted of 500 nM forward and reverse SARS-CoV-2 N TaqMan assay primers, 250 nM of each probe, 5 µL 2× TaqMan Fast Advanced Master Mix (Thermo-Fisher), and either 2.5 µL of plasmid standards at 0.4E2–0.4E7 plasmid copies/µL (final of 1E2 – 1E7 plasmid copies in reaction) or 2.5 µL of diluted cDNA sample. The cDNA from mouse samples was diluted 1:50. Reaction cycling conditions were as follows: *UNG activation at 50 °C for 2 min, initial denaturation at 95 °C for min, cycling at 40× (denaturation at 95 °C - 1 s, annealing and amplification at 60 °C - 20 s).* Probes were validated to be equally effective in detecting viral transcripts between singleplex and duplex reactions.

Resulting $C_q$ values were used to determine the number of viral transcript copies of a single viral species detected in the reaction by relating it against the corresponding standard curve in singleplex or duplex reactions. The total number of viral transcripts detected in the reaction was determined by taking the sum of detected viral transcript copies across both species. The percentage population of individual viral species within a reaction was expressed as a relative ratio to the total viral transcript copy number detected. TaqMan probe specificity was validated in RT-qPCR assays using cDNA from RNA isolated from recombinant SARS-CoV-2-infected HUVEC[ACE2] cells (S2I Fig).

## Virus propagation, titration, and infection

Experiments with SARS-CoV-2 variants and rSARS-2 were conducted in a CL3 facility, and all standard operating procedures were approved by the CL3 Operations Oversight Committee and Biosafety Office at the University of Calgary (viral propagation) or Université Laval. SARS-CoV-2 Toronto-01 (TO-1), Alpha, Beta, Gamma, and Delta isolates were titrated and propagated in VeroE6 cells, as in Kleer and colleagues [14]. Omicron BA-1 isolate (a generous gift from Dr. Lorne Tyrrell, Dr. Michael Joyce, and Holly Bandi [University of Alberta]) and all rSARS-2 were titrated and propagated in Vero-E6[TMPRSS2] cells. Virus infectious titer was enumerated by plaque assay on VeroE6 or VeroE6[TMPRSS2] (Omicron and rSARS-2) cells using equal parts 2.4% w/v semi-solid colloidal cellulose overlay (Sigma; prepared in ddH$_2$O) and 2× DMEM (Wisent) with 1% FBS, 100 U/mL penicillin, 100 µg/mL streptomycin, 2 mM L-glutamine. 48 to 72 hours post-infection cells were fixed with 4% formaldehyde and stained with 1% w/v crystal violet (Thermo-Fisher) for plaque enumeration. For propagation, cells were infected at an MOI of 0.01 for 1 hour in serum-free, antibiotic-free DMEM at 37 °C. After virus adsorption, media was replaced with fresh media, containing 2% FBS and 100 U/mL penicillin, 100 µg/mL streptomycin, 2 mM L-glutamine. 3–5 days post-infection, virus-containing supernatant was harvested and cell debris was removed by centrifugation. Virus stocks were stored at −80 °C and infectious titer was determined by plaque assay.

For experiments, cells were seeded to achieve ~80% confluence at the time of infection. Virus inoculum was diluted in serum-free, antibiotic-free DMEM to achieve the desired MOI. Growth media was removed from cells and replaced with the diluted virus inoculum. Cells were infected for 1 hour at 37 °C, after which the media was replaced with complete growth media until experimental endpoint.

## Extracellular virus particle concentration

VeroE6 cells were infected with SARS-CoV-2 TO-1 or Alpha (MOI = 1) in T75 cm$^2$ flasks. Twenty-four hpi, virus-containing supernatant was collected, and cell debris was removed by low-speed centrifugation (1,500 RPM for 5 min) and filter sterilization using a 0.45 µm filter (VWR). Mature viral particles were concentrated as in [104]. Briefly, viruses were concentrated by high-speed centrifugation (20,000 RPM for 4 hours; Beckman Coulter Optima LK-90, with T70.1 fixed-angle rotor) through a 20% (w/v in PBS) sucrose cushion. Viral pellet was resuspended in 2× Laemmli buffer and boiled at 92 °C for 15 min.

## Fluorescent RNA oligonucleotides

Complementary RNA oligonucleotides with the following sequences were designed as in [48] and purchased from Thermo-Fisher:

**38 nt:** 5′-AUGAAGGUUUGAGUUGAGUGGAGAUAGUGGAGGGUAGU-3′

**Fluorescein-18 nt:** 5′-fluorescein-CUCAACUCAAACCUUCAU-3′

 Oligonucleotides were resuspended in Nuclease-Free Duplex Buffer (30 mM HEPES, pH 7.5; 100 mM potassium acetate) (Integrated DNA Technologies). To obtain dsRNA, 38 nt and fluorescein-18 nt oligonucleotides were combined at an equimolar ratio for a final concentration of 100 μM, heated for 5 min at 95 °C, then gradually cooled to 4 °C. RNA oligonucleotide transfections were performed using Lipofectamine 2000 (see Materials and methods "Transient transfections and drug treatments").

### Reverse transcriptase quantitative and non-quantitative-PCR

Intracellular RNA was harvested using RNeasy Mini kit (Qiagen) according to manufacturer's instructions. Extracellular RNA was extracted using QIAamp Viral RNA Mini kit (Qiagen) as per manufacturer's instructions, without including carrier RNA. Extracted RNA was stored at −80 °C until use. RNA concentrations were determined using NanoDrop One$^C$ (Thermo-Fisher) and 1,000 ng of RNA was converted to cDNA using Maxima H Minus Reverse Transcriptase (Thermo-Fisher) according to manufacturer's instructions. For RT-PCR, following PCR amplification with Taq DNA polymerase (Thermo-Fisher), amplicons were resolved by agarose gel electrophoresis on a 1.5% gel with SYBR Safe (Thermo-Fisher). RT-PCR cycling conditions: initial denaturation at *95 °C - 30 s; cycling at 30 × (95 °C - 30 s; annealing at 59 °C - 30 s; extension at 68 °C - 5 s); final extension at 68 °C - 5 min.* Resolved amplicons were imaged using the Chemi-Doc MP Imaging System. For RT-qPCR, cDNA was diluted 1:10 with nuclease-free $H_2O$ and cDNA was amplified using SsoFast EvaGreen Master Mix (BioRad). RT-qPCR cycling conditions: initial denaturation at *98 °C - 2:00 min; cycling at 39 × (98 °C - 0.02 min; annealing and amplification at 60 °C - 0.05 min); final extension at 65 °C - 0.10 min, followed by melt curve analysis (60 °C - 0.10 min and 95 °C - 0.2 min).* Relative RNA abundance was calculated using the $2^{-\Delta\Delta Ct}$ equation, with 18S rRNA as a housekeeping control gene. Primer sequences can be found in Table 4.

### Transient transfections and drug treatments

Transient transfections in HeLas were conducted using Fugene HD (Promega) according to the manufacturer's guidelines. Briefly, in Opti-MEM (Gibco), plasmid DNA and Fugene HD were added, with 1 μg of DNA to 3 μL of Fugene HD. The transfection mixture was added dropwise to cells containing antibiotic-free media. Twenty-four hours post-transfection, the media was replaced. Poly I:C and poly A:U (Invitrogen) transfections were conducted with Fugene HD, as described above. RNA oligonucleotide transfections were conducted with Lipofectamine 2000 (Thermo-Fisher) according to the manufacturer's guidelines. In Opti-MEM, RNA oligonucleotides and Lipofectamine 2000 were added in a 1:1 ratio (1 μg of RNA to 1 μL of Lipofectamine 2000). Transfection mixture was added dropwise to cells containing antibiotic-free media. All other transfections were achieved using polyethylimine (PEI, Polysciences). In Opti-MEM, plasmid DNA and PEI (1 mg/mL) were added, with 1 μg of DNA to 3 μL of PEI. Transfection mixture was added dropwise to cells containing

**Table 4. RT-PCR and qPCR Primer sequences.**

| Target | Forward Sequence (5′ → 3′) | Reverse Sequence (5′ → 3′) |
|---|---|---|
| sgRNA-*N/N\** | CTTTCGATCTCTTGTAGATCTGTTC-TCTAAACGAAC | GTTCTGGACCACGTCTGCC |
| SARS2 *ORF1ab* Standard | TACGTGCATGGATTGGCTT | CAGCAACTAGGTTAACACCTGTA |
| SARS2 *N* TaqMan Assay | TCATCACGTAGTCGCAACAG | CAAAGCAAGAGCAGCATCAC |
| *18S* | TTCGAACGTCTGCCCTATCAA | GATGTGGTAGCCGTTTCTCAGG |
| *IFNβ* | GTCTCCTCCAAATTGCTCTC [105] | ACAGGAGCTTCTGACACTGA [105] |

serum-free, antibiotic-free media. 6 hours post-transfection, the media was replaced. Sodium arsenite (SA; Sigma) was added to cells at a 500 µM final concentration.

## Production and use of recombinant lentiviruses

Second-generation recombinant lentiviruses were produced as in Kleer and colleagues [14]. Briefly, HEK293T cells were transfected with pMD2.G, psPAX2, and the lentiviral transfer plasmid at a ratio of 1:2:3.3 (µg) using PEI as described above. 6 hours post-transfection, the media was replaced with DMEM containing 10% serum, but no antibiotics. Forty-eight hours post-transfection, virus supernatant was harvested, filtered with a 0.45 µm syringe filter (VWR), and stored at −80 °C for single use. For transductions, lentiviruses were thawed at 37 °C and added to target cells in complete media containing 5 µg/mL polybrene (Sigma). Twenty-four hours post-transduction, media was replaced with complete media containing puromycin (1 µg/mL; Thermo-Fisher) or blasticidin (5 µg/mL; Thermo-Fisher). Cells were selected for 48 hours before replacing with complete media or until experimental endpoint.

## Co-immunoprecipitations

Co-immunoprecipitations were conducted as in [106]. Briefly, HEK293T cells were seeded into a 6-well plate. Co-immunoprecipitations were conducted 48 hours post-plasmid transfection. Cells were transfected with poly I:C or mock-transfected 3 hours before conducting co-immunoprecipitation. Cells were resuspended in lysis buffer (150 mM NaCl, 10 mM Tris pH 7.4, 1 mM EDTA, 1% v/v Triton X-100, 0.5% v/v NP-40, Roche protease inhibitor tablet). For Fig 5A, 10 or 50 µg/mL of RNase A (Thermo-Fisher) was added to lysis buffer and incubated for 30 min at room temperature. Following lysis, cells were incubated with the indicated primary antibody overnight at 4 °C. Protein G magnetic Dynabeads (Thermo-Fisher) were incubated in blocking buffer (150 mM NaCl, 10 mM Tris pH 7.4, 1 mM EDTA, 5 mg/mL BSA) overnight at 4 °C. Immunoprecipi-tation was conducted and beads were boiled in 4× Laemmli buffer (BioRad) with 10% v/v β-mercaptoethanol.

## Immunoblotting and densitometric analysis

Cells were lysed in 2× Laemmli buffer and stored at −20 °C until use. For SARS-CoV-2-infected cells, cells were lysed in 2× Laemmli buffer and boiled at 92 °C for 15 min, then stored at −20 °C. Protein concentration was quantified for nor-malization using DC Protein Assay kit (BioRad) according to manufacturer's instructions. 6–15 µg of protein lysate was resolved by SDS-PAGE using TGX Stain-Free acrylamide gels (BioRad). After protein transfer, total protein images were acquired from PVDF membranes on the ChemiDoc MP Imaging System (BioRad). PVDF membranes were blocked with 5% BSA (Thermo-Fisher) in TBST (Tris-buffered saline solution with 0.1% Tween-20). Primary antibodies and second-ary antibodies were diluted in 5% BSA in TBST and 5% skim milk in TBST, respectively, except when probing for pIRF3, where the secondary antibody was diluted in 5% BSA. Antibody concentrations can be found in Table 5. Densitometry was conducted in ImageLab software (BioRad) by normalizing to total protein.

## Poly I:C-biotin pulldowns

The poly I:C-biotin pulldown protocol was adapted from [107]. HEK293T cells were seeded into a 6-well plate. Lysates were collected 24 hours post-plasmid transfection. The media was removed and cells were washed with ice-cold 1× PBS, then harvested in ice-cold 1× PBS and pelleted by centrifugation. Cells were then resuspended in ice-cold 1× PBS, pelleted by centrifugation, and resuspended in lysis buffer (150 mM NaCl, 10 mM Tris pH 7.4, 1 mM EDTA, 1% v/v Triton X-100, 0.5% v/v NP-40, 0.1% v/v P8340 protease inhibitor cocktail (Sigma Aldrich)). Cell lysates were combined with 1 U/µL RNase T1 (Thermo-Fisher) and rocked for 30 min at room temperature, then rotated at 4 °C for 30 min and subjected to centrifugation to remove cell debris.

 M270 Streptavidin magnetic Dynabeads (Thermo-Fisher) were washed three times with 1× binding and washing (B&W) buffer (2 M NaCl, 1 mM EDTA, 10 mM Tris-HCl pH 7.4). Beads were combined with 1× B&W buffer containing 300 ng

**Table 5. Antibodies.**

| Antibody | Species | Vendor/Catalog # | Application | Dilution |
|---|---|---|---|---|
| ACE2 | Rabbit | CST/4355S | Immunoblot | 1:1,000 |
| Actin-HRP conjugated | Rabbit | CST/5125S | Immunoblot | 1:10,000 |
| FLAG | Chicken | Abcam/ab1170 | Immunofluorescence | 1:100 |
| FLAG | Mouse | CST/8146S | Immunoblot Immunoprecipitation | 1:1,000 1:1,000 |
| FLAG | Rabbit | CST/14793T | Immunoblot Immunofluorescence | 1:1,000 1:1,000 |
| G3BP1 | Mouse | BD/611126 | Immunoblot Immunofluorescence | 1:1,000 1:1,000 |
| HedIs | Mouse | Santa Cruz/8418 | Immunofluorescence | 1:1,000 |
| HA | Rabbit | CST/3724S | Immunofluorescence | 1:1,000 |
| IRF3 | Rabbit | CST/11904T | Immunoblot | 1:1,000 |
| pIRF3 | Rabbit | CST/37829 | Immunoblot | 1:500 |
| J2 | Mouse | Millipore/MABE1134 | Immunofluorescence | 1:100 |
| J2 (the above was discontinued) | Mouse | CST/76651L | Immunofluorescence | 1:500 |
| PKR | Rabbit | Abcam/ab184257 | Immunoblot | 1:1,000 |
| pPKR | Rabbit | Abcam/ab32036 | Immunoblot | 1:1,000 |
| SARS-CoV-2 N | Rabbit | Novus/NBP3–05730 | Immunoblot Immunofluorescence | 1:1,000 1:1,000 |
| TIAR | Rabbit | CST/8509T | Immunofluorescence | 1:800 |
| Alexa Fluor 405 anti-rabbit IgG | Donkey | Abcam/ab175651 | Immunofluorescence | 1:200 |
| Alexa Fluor 488 anti-rabbit IgG | Chicken | Invitrogen (A21441) | Immunofluorescence | 1:1,000 |
| Alexa Fluor 488 anti-mouse IgG | Goat | Invitrogen (A11029) | Immunofluorescence | 1:1,000 |
| Alexa Fluor 555 anti-mouse IgG | Donkey | Invitrogen (A31570) | Immunofluorescence | 1:1,000 |
| Alexa Fluor 647 anti-mouse IgG | Chicken | Invitrogen (A21463) | Immunofluorescence | 1:1,000 |
| Alexa Fluor 488 anti-chicken IgY | Goat | Invitrogen (A11039) | Immunofluorescence | 1:200 |

poly I:C (HMW)-biotin (Invivogen) and 200 U/mL RNaseOUT Recombinant Ribonuclease Inhibitor (Thermo-Fisher) and rotated at 4 °C for 2.5 hours, then washed three times with 1× B&W buffer containing 200 U/mL RNaseOUT Recombinant Ribonuclease Inhibitor. Cell lysates were combined with beads and rotated at 4 °C overnight. Beads were washed three times with wash buffer (150 mM NaCl, 10 mM Tris pH 7.4, 1 mM EDTA), then elution was performed by boiling beads in 4× Laemmli buffer (BioRad) with 10% v/v β-mercaptoethanol.

## rRNA integrity assay

Cells were transfected with 1.0 μg high molecular weight poly I:C (Invitrogen) for 3 hours. Intracellular RNA was harvested using RNeasy Mini kit (Qiagen) according to manufacturer's instructions. Extracted RNA was stored at −80 °C until use. RNA concentrations were determined using NanoDrop One$^C$ (Thermo-Fisher). Extracted RNA was diluted to a final concentration of 50 ng/μL, then submitted to the University of Calgary Centre for Health Genomics and Informatics for automated electrophoresis using an Agilent 4200 TapeStation system.

## Immunofluorescence

Cells were seeded onto 18 mm glass round-bottom #1.5 coverslips (Electron Microscopy Sciences). At the experimental endpoint, cells were fixed with 4% paraformaldehyde (Electron Microscopy Sciences) in PBS (Gibco) for 10 min at room temperature, or 30 min if infected with SARS-CoV-2. Cells were permeabilized with 0.1% (v/v) Triton X-100

(Sigma-Aldrich) in 1× PBS for 10 min at room temperature. Cells were blocked with 1% (v/v) human AB serum (Sigma-Aldrich) in PBS for 60 min at room temperature. Primary and secondary antibodies were diluted in 1% human AB blocking buffer at listed concentration (Table 5). Primary antibodies were incubated overnight at 4 °C, secondary antibodies were incubated for 60 min at room temperature. Nuclei were stained with 1 μg/mL Hoechst (Invitrogen). Coverslips were mounted with antifade Prolong Gold (Thermo-Fisher). Unless otherwise stated, cells were co-stained with Alexa Fluor 647 and Alexa Fluor 488 to minimize fluorophore cross-excitation and bleed-through. Images were captured using a Zeiss AxioObserver Z1 microscope with a 40× oil-immersion objective for all images used for G3BP1 foci and P-body quantification. For co-localization analysis and all representative images, images were acquired using a Zeiss LSM 880 confocal microscope using a 63× oil-immersion objective. Exposure time and laser power were kept consistent within replicates.

### RNP granule quantification and enrichment analysis

P-bodies and G3BP1 foci were quantified using CellProfiler4.0.6 (cellprofiler.org), as in Kleer and colleagues [14]. Nuclear staining was used to identify individual cells by applying manual thresholding followed by "Identify Primary Objects". Acceptable nuclei size was cell-type specific and dependent on imaging objective. Cell borders were defined by applying a "Propagation" function from each nucleus. Propagation distance was dependent on imaging objective and was cell-type specific to account for differences in cell size. Nuclei in close proximity have automatically reduced propagation distance, such that each cytoplasm is mutually exclusive and therefore cytoplasmic area or granules are not double-counted. As P-bodies and G3BP1 foci are cytoplasmic, the nuclear area was subtracted using "Mask" and "Remove" functions. The cytoplasmic area was masked and RNP granules were measured within each cell. Cells were then stratified into "Positive Cells" and "All Cells." Positive Cells are those that stain positive for N (virus infection) or FLAG (over-expression). "All Cells" include cells expressing N/FLAG and non-expressing cells. In this way, in control treatments (i.e., mock-infected cells or EV-transfected cells), RNP granules in All Cells are calculated,whereas, in experimental treatments (i.e., infected cells or N-transfected cells), RNP granules in only Positive Cells are calculated. To identify RNP granules, granules were thresholded to reduce background staining. Granules were defined based on size; P-bodies were defined as Hedls-positive puncta ranging from 3 to 13 pixels and SGs were defined as G3BP1-positive puncta ranging from 2 to 50 pixels for poly I:C-induced G3BP1 foci and 3–50 pixels for SA-induced G3BP1 foci. Quantification parameters were identical within independent experiments; however, image thresholding was altered between independent experiments to account for staining variability. P-bodies and G3BP1 foci were quantified and graphed as number of foci per cell.

For enrichment of N proteoforms in dsRNA foci, images were all acquired on a Zeiss LSM 880 confocal microscope with z-stacks. Nuclei and cell borders were defined as above; however, as dsRNA foci can be found overlapping with nuclei, the nucleus was not subtracted from the total cellular area. Nucleocapsid staining was thresholded and N positive cells were identified as above. dsRNA foci were identified by thresholding and size exclusion as above. To measure nucleocapsid localization inside dsRNA foci, puncta were subjected to a "Mask" and "Keep" function. To measure nucleocapsid localization outside of SGs or dsRNA foci, puncta were subjected to a "Mask" and "Remove" function. The mean integrated intensity of unthresholded N staining from "Nucleocapsid Inside Puncta" and "Nucleocapsid Outside Puncta" were measured. Percent enrichment of N in puncta was calculated by addition of the mean integrated intensity of N Inside Puncta divided by the total cellular integrated intensity (integrated intensity of "Nucleocapsid Inside Puncta" plus integrated intensity of "Nucleocapsid Outside Puncta"). N-puncta enrichment was calculated and represented per cell. Quantification parameters were identical within independent experiments; however, image thresholding was altered between independent experiments to account for staining variability. For Fig 8, images were blinded before G3BP1 foci and TIAR foci were quantified manually.

### SARS-CoV-2 genomic surveillance

SARS-CoV-2 sequences and associated metadata were acquired from Nextstrain.org (https://nextstrain.org/ncov/open/global/all-time). This is a curated dataset sourced from publicly available genome data between late 2019 and September

22, 2025. The aligned sequences (aligned.fasta) and the associated metadata (metadata.tsv) were downloaded from Nextstrain.org. All processing was conducted in RStudio (version 2025.05.1 + 513) unless otherwise indicated. The following programs were used in R: seqKit and ggplot2. The metadata was filtered to retain only the relevant information for our analysis, including "date," "pango_lineage," "substitutions," and "genbank_accession." Samples with missing or incomplete collection date entries were excluded from further analysis (starting genomes: 9,383,761; after filtering: 9,383,034).

The grepl() function was used to track the prevalence of specific mutations (G28881A/G28882A/G28883C; and G28884C). In the case of the G28881A/G28882A/G28883C mutation, only genomes with all three mutations were considered positive. The daily prevalence of the mutation was calculated by dividing the number of mutation-positive genomes by the total number of genomes corresponding to the same day. The limited sequences collected in late 2019 and early 2020 resulted in an imbalanced dataset which was disproportionately volatile (S6A and S6B Fig). To balance the dataset, only days where more than 30 genomes were collected were included in the analysis. Analysis of all days with SARS-CoV-2 sequences collected revealed large fluctuations in late 2019 and early 2020.

To identify the prevalence of the N* TRS, a data frame was constructed to link the filtered metadata and the aligned genomes. Briefly, to generate a temporally balanced dataset, a maximum of 2000 genomes were randomly sampled for each year between 2019 and 2025. Using terminal-based commands on a macOS, the sampled metadata was coupled to the aligned genomes using the accession number. With the data frame built, the presence of the TRS within the N gene (N* TRS) was examined by isolating the aligned N genes (nucleotides 28274-29533) and using the grel() function for the TRS motif (ACGAAC). The weekly prevalence of the N* TRS was calculated by dividing the number of N* TRS-positive genomes by the total number of genomes corresponding to the same week. Pipelines are available on github https://github.com/rmulloy97/Mulloy-et-al.-SARS-CoV-2-TRS-prevalence or Zenodo.org DOI: https://doi.org/10.5281/zenodo.18225476.

## Statistics

All statistical analyses were performed using GraphPad Prism 9.0 or 10.0. RNP granule (G3BP1 foci and P-bodies) per cell foci counts as well as per cell enrichment quantifications were plotted such that independent biological replicates were combined and plotted on one graph. These per cell data are naturally skewed and non-parametric (as determined by the Shapiro-Wilk normality test); therefore, we used rank-sum statistical analyses (Mann–Whitney $U$ test for comparing two independent groups, and Kruskal–Wallis test for comparing multiple independent groups). All other statistics were performed using parametric statistical analyses ($T$ test for comparing two independent groups or ANOVA for comparing multiple independent groups) unless otherwise indicated.

## Supporting information

**S1 Fig. Methionine 210 and 234 enable truncated N production. A–D.** Protein quantification from Fig 2F (A and B) and H (C and D) was conducted by densitometry. N* proteoform abundance is presented relative to $N^{FL}$ abundance ($N^*/N^{FL}$). These data represent two independent biological replicates ($n = 2$), SD. See S1 Data for densitometry values. **E.** $N^{*M210-WT}$, or $N^{*M210-M234V}$ containing a C-terminal FLAG tag, or an empty vector (EV) control were over-expressed in HEK293T cells and protein lysate was subjected to SDS-PAGE and immunoblotting with anti-FLAG and anti-actin antibodies. See S7 Supporting Data for full blots. **F.** $N^{*M210-WT}$, or $N^{FL-M210I/M234V}$ containing a C-terminal FLAG tag, or an empty vector (EV) control were over-expressed in HEK293T cells and protein lysate was subjected to SDS-PAGE and immunoblotting with anti-FLAG and anti-actin antibodies. See S7 Supporting Data for full blots. **G.** Kinetics of N proteoform and sgRNA profile was determined by infecting Calu3 cells with SARS-CoV-2 Alpha variant (MOI = 4). At the indicated time post-infection, RNA and protein lysate was harvested and was subjected to RT-PCR and agarose gel electrophoresis, and SDS-PAGE and immunoblotting, respectively. See S7 Supporting Data for full blots.
(S1_Fig.TIFF)

**S2 Fig. rSARS-CoV-2 validation and additional co-infection competition assay analysis. A.** sgRNA profile of recombinant SARS-CoV-2 was determined by infecting HUVEC$^{ACE2}$ cells (MOI = 4). RNA was harvested 24 hpi and subjected to RT-PCR and agarose gel electrophoresis. See S8 Supporting Data for full gel. **B, C.** Protein quantification from Fig 3C was conducted by densitometry. N* proteoform abundance is presented relative to N$^{FL}$ abundance (N*/N$^{FL}$). See S1 Data for densitometry values. **D.** N proteoform profile of SARS-CoV-2 variants was determined by infecting HUVEC$^{ACE2}$ cells (MOI = 4). Protein lysate was harvested 24 hpi and subjected to SDS-PAGE and immunoblotting with anti-N. See S8 Supporting Data for full blots. **E–G.** Protein quantification from S2D Fig was conducted by densitometry. N* proteoform abundance is presented relative to N$^{FL}$ abundance (N*/N$^{FL}$). See S1 Data for densitometry values. **H.** Schematic of DNA probes used to differentiate rSARS-CoV-2 viruses. Probe and respective target sequence to either Site 1 (to differentiate WT, KR$^{+TRS}$, and KR$^{-TRS}$) or Site 2 (to differentiate WT from M210I). **I.** TaqMan probe specificity was validated in RT-qPCR assays using cDNA as a template, generated from RNA from rSARS-CoV-2-infected HUVEC$^{ACE2}$ cells (24 hpi, MOI = 4). All possible combinations of probes and recombinant virus-infected samples were tested alongside a no-template control (NTC); resulting $C_q$ values are displayed on a heatmap where reactions without amplification were set to $C_q$ = 40. See S1 Data for quantification values. **J–L.** Primary HUVEC$^{ACE2}$ cells were coinfected with equal infectious titers of the indicated recombinant virus to achieve a total MOI of 0.02. Time 0 represents the inferred proportions of each recombinant based on infectious titer input. At the indicated time post-infection, virus-containing supernatant was harvested, cell debris was removed by centrifugation (5 min at 1,000 RPMs), and extracellular RNA was harvested and subjected to probe-based RT-qPCR to differentiate recombinant virus abundance. These data represent three independent biological replicates (n = 3). Statistics were performed using ratio paired T test (* $p < 0.0332$, ** $p < 0.0021$), standard error mean; SEM. See S1 Data for quantification values. **M.** Infectious titer from K18$^{ACE2}$ mice co-infected with KR$^{+TRS}$ and KR$^{-TRS}$ from Fig 3M was determined using lung tissue homogenate and enumerated by TCID$_{50}$ at day 1 post-infection (n = 4) and day 3 post-infection (n = 5), standard error mean; SEM. See S1 Data for quantification values. (S2_Fig.TIFF)

**S3 Fig. N*$^{M210}$ is superior at sequestering dsRNA compared to N$^{FL}$. A.** HeLa cells were transfected with EV, mock-transfected (treated with Fugene HD only), or not treated. Twenty-four hours post-transfection, cells were fixed, and the nuclei were stained with Hoechst. A maximum intensity projection is presented. Scale bar = 10 μm. **B.** A549 cells were transduced with recombinant lentiviruses to ectopically express N$^{FL+/-dsRBM}$ or N*$^{M210+/-dsRBM}$ or EV. For all N proteoforms, internal methionine residues were mutated to ensure only the indicated proteoform of N was expressed ([N$^{FL}$; M210I and M234V], [N*$^{M210}$; M234V]). Ninety-six hours post-transduction, cells were transfected with 0.5 μg high molecular weight poly I:C or mock-transfected. Three hours post-transfection, cells were fixed and immunostained with the FLAG antibody (N proteoform; Alexa 488) and J2 antibody (dsRNA; Alexa 647). Nuclei were stained with Hoechst. A maximum intensity projection is presented here. One representative experiment of two independent replicates is shown. Scale bar = 10 μm. **C.** Enrichment of N proteoforms with RNA as in Fig 4D. These data represent two independent biological replicates (n = 2) with 20 cells measured per condition, per replicate. See S1 Data for quantification values. (S3_Fig.TIFF)

**S4 Fig. Compared N$^{FL}$, N*$^{M210}$ more readily co-localizes with dsRNA. A.** EV, N*$^{M210}$, or N*$^{M210-ΔdsRBM}$-expressing HeLa cells were transfected with 0.5 μg of high molecular weight (HMW), low molecular weight (LMW) poly I:C, or poly A:U, or mock-transfected. Three hours post-transfection, cells were fixed and immunostained with the FLAG antibody (N proteoform; Alexa 488) and J2 antibody (dsRNA; Alexa 647). Nuclei were stained with Hoechst. A maximum intensity projection (MIP) is presented here. One representative experiment of three independent replicates is shown (n = 3). Scale bar = 10 μm. **B.** Enrichment of N proteoforms with dsRNA as in Fig 4D. These data represent three independent biological replicates (n = 3) with 18 cells measured per condition, per replicate. Statistics were performed using a Mann-Whitney test (*** $p < 0.0002$, **** $p < 0.0001$). See S1 Data for quantification values. (S4_Fig.TIFF)

**S5 Fig. The dsRBM of N\*M210 is required for P-body disassembly. A.** Primary HUVEC cells were transduced with recombinant lentiviruses to ectopically express $N^{FL+/-dsRBM}$, $N^{*M210+/-dsRBM}$, with internal methionines mutated ([$N^{FL}$; M210I and M234V], [$N^{*M210}$; M234V]), or EV. Ninety-six hours post-transduction, cells were fixed and immunostained with the FLAG antibody (N proteoform; Alexa 488) and the Hedls antibody (P-bodies; Alexa 647). Nuclei were stained with Hoechst. A maximum intensity projection (MIP) is presented here. Scale bar = 10 μm. **B.** P-bodies were quantified using CellProfiler by measuring Hedls puncta in N-expressing cells (thresholded by FLAG staining) or EV transduced cells. These data represent three independent biological replicates ($n = 3$) with >20 cells measured per condition, per replicate. Each data-point represents a single cell. Statistics were performed using a Kruskal–Wallis $H$ test with Dunn's correction (* $p < 0.032$, ** $p < 0.0021$, **** $p < 0.0001$). See S1 Data for quantification values. **C, D.** Quantification of G3BP1 (C) or TIAR (D) foci from all N-positive cells (combined high and low expressors) from Fig 8A–8E. Foci were quantified using CellProfiler by enumerating G3BP1 or TIAR foci in N-expressing cells. Each datapoint represents a single cell with 100 cells measured per condition. Statistics were performed using an unpaired $T$ test; mean; SEM. See S1 Data for quantification values. **E.** Protein lysates from wild-type and G3BP1 knockout $HEK293A^{ACE2}$ (ΔG3BP1) cells were harvested and resolved by SDS-PAGE and immunoblotted with anti-G3BP1 and anti-ACE2 antibodies. See S9 Supporting Data f or full blots.
(S5_Fig.TIFF)

**S6 Fig. Filtering SARS-CoV-2 sequences for N\* TRS analysis. A.** The prevalence of the G28881A/G28882A/G28883C mutation was determined as in Fig 9C and the genomes with the G28881A/G28882A/G28883C mutation were plotted over time. Here, sequences collected from all days are plotted, not just days with >30 sequences collected per day as in Fig 9C. 28882A and 28883C are highlighted in red as they contributed to the N\* TRS. **B.** The number of sequences collected per day were extracted from Nextstrain.org showing the low sequencing coverage at the end of 2019 and start of 2020.
(S6_Fig.TIFF)

**S1 Data. Source Data.**
(S1_Data.XLSX)

**S1 Supporting Data. A and B.** Agarose gel showing sgRNA-N\* synthesis from SARS-CoV-2 variant infections. **A.** Lane 1 = mock-infected, Lane 2 = TO-1, Lane 3 = Alpha, Lane 4 = Beta, Lane 5 = Gamma, Lane 6 = Delta. **B.** Lane 1 = mock-infected, Lane 2 = TO-1, Lane 3 = Omicron BA1. **C–E.** Immunoblot showing the synthesis of N\* proteoforms from SARS-CoV-2 variant infections. **C.** Lane 1 = mock-infected, Lane 2 = TO-1, Lane 3 = Alpha, Lane 4 = Beta, Lane 5 = Gamma, Lane 6 = Delta. **D.** Lane 1 = mock-infected, Lane 2 = TO-1, Lane 3 = Omicron BA1. **E.** And the presence of N\* in isolated virus particles; Lane 1 = mock-infected, Lane 2 = TO-1, Lane 3 = Alpha. **F.** $N^{FL}$ over-expression with various methionine muta-tions. Lane 1 = EV, Lane 2 = $N^{FL-WT}$, Lane 3 = $N^{FL-M101G}$, Lane 4 = $N^{FL-M210T}$, Lane 5 = $N^{FL-M210I}$, Lane 6 = $N^{FL-M234V}$.
(S1_Supporting Data.TIFF)

**S2 Supporting Data. A.** Agarose gel showing sgRNA-N\* synthesis from recombinant SARS-CoV-2 following infection of $A549^{ACE2}$ cells. A. Lane 1 = mock-infected, Lane 2 = WT, Lane 3 = $KR^{+TRS}$, Lane 4 = $KR^{-TRS}$, Lane 5 = M210I. **B.** Immuno-blot showing the synthesis of N\* proteoforms from recombinant SARS-CoV-2 following infection of $A549^{ACE2}$ cells. Lane 1 = mock-infected, Lane 2 = WT, Lane 3 = $KR^{+TRS}$, Lane 4 = $KR^{-TRS}$, Lane 5 = M210I.
(S2_Supporting Data.TIFF)

**S3 Supporting Data. A.** Immunoblots from poly I:C immuno-precipitation. Lanes 1–4 = input (Lane 1 = EV, Lane 2 = $N^{FL}$, Lane 3 = $N^{*M210}$, Lane 4 = $N^{*M210-ΔdsRBM}$); Lanes 5–8 = pulldown with no poly I:C (Lane 5 = EV, Lane 6 = $N^{FL}$, Lane 7 = $N^{*M210}$, Lane 8 = $N^{*M210-ΔdsRBM}$); Lanes 9–12 = pulldown with poly I:C (Lane 9 = EV, Lane 10 = $N^{FL}$, Lane 11 = $N^{*M210}$, Lane 12 = $N^{*M210-ΔdsRBM}$).
(S3_Supporting Data.TIFF)

**S4 Supporting Data. A.** Immunoblot from $N^{FL\text{-}WT\text{-}HA}$:$N^{*M210\text{-}FLAG}$ co-IP. Lanes 1–6 = input (Lane 1 = EV, Lane 2 = $N^{FL\text{-}WT\text{-}HA}$, Lane 3 = $N^{*M210\text{-}FLAG}$, Lane 4 = $N^{FL\text{-}WT\text{-}HA}$:$N^{*M210\text{-}HA}$, Lane 5 = $N^{FL\text{-}WT\text{-}HA}$:$N^{*M210\text{-}HA}$ with 10 µg/mL RNase A, Lane 6 = $N^{FL\text{-}WT\text{-}HA}$:$N^{*M210\text{-}HA}$ with 50 µg/mL RNase A); Lanes 7–12 = IP (Lane 7 = EV, Lane 8 = $N^{FL\text{-}WT\text{-}HA}$, Lane 9 = $N^{*M210\text{-}FLAG}$, Lane 10 = $N^{FL\text{-}WT\text{-}HA}$:$N^{*M210\text{-}HA}$, Lane 11 = $N^{FL\text{-}WT\text{-}HA}$:$N^{*M210\text{-}HA}$ with 10 µg/mL RNase A, Lane 12 = $N^{FL\text{-}WT\text{-}HA}$:$N^{*M210\text{-}HA}$ with 50 µg/mL RNase A). **B.** Co-expression of $N^{FL\text{-}FLAG}$ and $N^{*M210\text{-}HA}$. Lane 1 = EV, Lane 2 = $N^{FL\text{-}FLAG}$, Lane 3 = $N^{FL\text{-}FLAG}$:$N^{*M210\text{-}HA}$ [8:1], Lane 4 = $N^{FL\text{-}FLAG}$:$N^{*M210\text{-}HA}$ [2:1].
(S4_Supporting Data.TIFF)

**S5 Supporting Data. A.** Immunoblots showing $N^{FL}$/$N^{*M210}$ inhibition of poly I:C-induced immune responses. Lanes 1–3 = mock-treated (Lane 1 = EV, Lane 2 = NFL, Lane 3 = $N^{*M210}$); Lanes 4–6 = poly I:C-treated (Lane 4 = EV, Lane 5 = NFL, Lane 6 = $N^{*M210}$). **B.** Immunoblots showing $N^{*M210}$ inhibition of poly I:C-induced immune responses. Lanes 1–3 = mock-treated (Lane 1 = EV, Lane 2 = $N^{*M210}$, Lane 3 = $N^{*M210\text{-}\Delta dsRBM}$); Lanes 4–6 = poly I:C-treated (Lane 4 = EV, Lane 5 = $N^{*M210}$, Lane 6 = $N^{*M210\text{-}\Delta dsRBM}$). **C.** TapeStation rRNA integrity automated electrophoresis. Lanes 1–3 = mock-treated (Lane 1 = EV, Lane 2 = NFL, Lane 3 = $N^{*M210}$); Lanes 4–6 = poly I:C-treated (Lane 4 = EV, Lane 5 = $N^{FL}$, Lane 6 = $N^{*M210}$). **D.** TapeStation rRNA integrity automated electrophoresis. Lanes 1–3 = mock-treated (Lane 1 = EV, Lane 2 = $N^{*M210}$, Lane 3 = $N^{*M210\text{-}\Delta dsRBM}$); Lanes 4–6 = poly I:C-treated (Lane 4 = EV, Lane 5 = $N^{*M210}$, Lane 6 = $N^{*M210\text{-}\Delta dsRBM}$). **E.** Immunoblot showing $KR^{+TRS}$ and $KR^{-TRS}$ immune activation. Lane 1 = mock-infected, Lane 2 = $KR^{+TRS}$, Lane 3 = $KR^{-TRS}$.
(S5_Supporting Data.TIFF)

**S6 Supporting Data. A.** Immunoblot showing $N^{FL}$-G3BP1 co-immunoprecipitation. Lanes 1–3 = mock-treated (Lane 1 = EV, Lane 2 = $N^{*M210}$, Lane 3 = $N^{FL}$); Lanes 4–6 = poly I:C-treated (Lane 4 = EV, Lane 5 = $N^{*M210}$, Lane 6 = $N^{FL}$).
(S6_Supporting Data.TIFF)

**S7 Supporting Data. A.** Lane 1 = EV, Lane 2 = N*M210-WT, Lane 3 = $N^{*M210\text{-}M234V}$. **B.** Lane 1 = EV, Lane 2 = $N^{FL\text{-}WT}$, Lane 3 = $N^{FL\text{-}M210I/M234V}$. **C.** Agarose gel and immunoblot of sgRNA-N* and $N^{*M210}$ synthesis kinetics. Lane 1 = 0 hpi, Lane 2 = 1 hpi, Lane 3 = 2 hpi, Lane 4 = 4 hpi, Lane 5 = 8 hpi, Lane 6 = 24 hpi.
(S7_Supporting Data.TIFF)

**S8 Supporting Data. A.** Agarose gel showing sgRNA-N* synthesis from recombinant SARS-CoV-2 following infection of $HUVEC^{ACE2}$ cells. A. Lane 1 = mock-infected, Lane 2 = WT, Lane 3 = $KR^{+TRS}$, Lane 4 = $KR^{-TRS}$, Lane 5 = M210I. **B.** Immunoblot showing the synthesis of N* proteoforms from recombinant SARS-CoV-2 following infection of $HUVEC^{ACE2}$ cells. Lane 1 = mock-infected, Lane 2 = WT, Lane 3 = $KR^{+TRS}$, Lane 4 = $KR^{-TRS}$, Lane 5 = M210I.
(S8_Supporting Data.TIFF)

**S9 Supporting Data. A.** Immunoblot validation of G3BP1 expression. Lane 1 = wild-type $HEK293A^{ACE2}$, Lane 2 = $HEK293A^{ACE2\text{-}\Delta G3BP1}$.
(S9_Supporting Data.TIFF)

## Acknowledgments

We sincerely thank Dr. Denys Khaperskyy (Dalhousie University) and Dr. James Burke (University of Florida Scripps Institute) for providing HEK293T-ΔG3BP1 cells and A549-ΔG3BP1/2 cells, respectively and for helpful discussions about this work. We would also like to thank Dr. Roy Duncan (Dalhousie University), Dr. Craig McCormick (Dalhousie University), Dr. James Burke (University of Florida Scripps Institute), and Dr. Christine Roden (Université de Montréal) for insightful comments about this work and Dr. Luis Martinez-Sobrido and Dr. Chengjin Ye (Texas Biomedical Research Institute) for the SARS-CoV-2 BACmid system and helpful discussions regarding recombinant SARS-CoV-2 cloning. We would also like to thank Sean Nesdoly (University of Calgary) for helpful conversations about bioinformatic analysis. This work would

not have been possible without Drs. Anne Vaahtokari and Luc Provencher of the Charbonneau Microscopy Facility and Drs. Devender Kumar and Shaunna Huston of the University of Calgary for CL3 support. We would like to thank Ryan Hisner for his detailed tracking and thoughtful commentary on SARS-CoV-2 evolution. Finally, we would like to thank all the members of the Corcoran lab, particularly Dr. Mariel Kleer, for helpful discussions about experimental design. All graphical schematics were made with Affinity Designer.

## Author contributions

**Conceptualization:** Jennifer A. Corcoran.

**Data curation:** Rory P. Mulloy.

**Formal analysis:** Rory P. Mulloy, Danyel Evseev, Noga Sharlin, Maxwell P. Bui-Marinos.

**Funding acquisition:** Jennifer A. Corcoran.

**Investigation:** Rory P. Mulloy, Danyel Evseev, Noga Sharlin, Maxwell P. Bui-Marinos, Émile Lacasse, Isabelle Dubuc.

**Methodology:** Rory P. Mulloy, Danyel Evseev, Noga Sharlin, Maxwell P. Bui-Marinos, Louis Flamand.

**Project administration:** Louis Flamand, Jennifer A. Corcoran.

**Resources:** Jennifer A. Corcoran.

**Supervision:** Jennifer A. Corcoran.

**Writing – original draft:** Rory P. Mulloy, Jennifer A. Corcoran.

**Writing – review & editing:** Rory P. Mulloy, Danyel Evseev, Noga Sharlin, Maxwell P. Bui-Marinos, Jennifer A. Corcoran.

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
