## [Editor Report · Decision Letter 0]

13 Jul 2025

Dear Dr Corcoran,

Thank you for submitting your manuscript entitled "A truncated SARS-CoV-2 nucleocapsid protein enhances virus fitness by evading antiviral responses" for consideration as a Research Article by PLOS Biology.

Your manuscript has now been evaluated by the PLOS Biology editorial staff, as well as by an academic editor with relevant expertise, and I am writing to let you know that we would like to send your submission out for external peer review under our anti-scooping policy regarding 10.1371/journal.pbio.3002982.

Once your full submission is complete, your paper will undergo a series of checks in preparation for peer review. After your manuscript has passed the checks it will be sent out for review. To provide the metadata for your submission, please Login to Editorial Manager (https://www.editorialmanager.com/pbiology) within two working days, i.e. by Jul 15 2025 11:59PM.

Kind regards,

Melissa

Melissa Vazquez Hernandez, Ph.D.

Associate Editor

PLOS Biology

---

## [Decision Letter · Decision Letter 1]

15 Aug 2025

Dear Dr Corcoran,

Thank you for your patience while your manuscript "A truncated SARS-CoV-2 nucleocapsid protein enhances virus fitness by evading antiviral responses" was peer-reviewed at PLOS Biology. It has now been evaluated by the PLOS Biology editors, an Academic Editor with relevant expertise, and by two independent reviewers.

In light of the reviews, which you will find at the end of this email, we would like to invite you to revise the work to thoroughly address the reviewers' reports. As you will see below, thre reviewers are positive about the study, yet some concerns have been raised during revision. Reviewer 1 thinks that two of the main conclusions are well supported but not the one related to the stress granule inhibition by N*M210 which may require a modification of the conclusion, or additional mechanistical data. The reviewer also mentions possible contamination in some IF images. Reviewer 2 thinks that while the study advances the field, most experiments rely on overexpression and the authors should therefore provide viral infection models. Specifically, the reviewer requires to show in infected cells the inhibition of stress granules, to assess if translation is differentially affected during infection, and that N*M210 inhibits polyI:C-induced SG formation in an infection model. We agree with all reviewer concerns and would require some additional experimental revisions to address them, as we consider that this would strengthen the work.

IMPORTANT: While R1 offers to clarify any points, please do so through us instead of contacting the reviewer directly.

Given the extent of revision needed, we cannot make a decision about publication until we have seen the revised manuscript and your response to the reviewers' comments. Your revised manuscript is likely to be sent for further evaluation by all or a subset of the reviewers.

**IMPORTANT - SUBMITTING YOUR REVISION**

*Re-submission Checklist*

*Published Peer Review*

*PLOS Data Policy*

*Blot and Gel Data Policy*

Sincerely,

Melissa

Melissa Vazquez Hernandez, Ph.D.

Associate Editor

PLOS Biology

REVIEWERS' COMMENTS:

Reviewer #1:

This manuscript examines the function of a SARS-2 NP variant synthesized from a novel sgRNA species. The main conclusion is that SARS-2 variants synthesize a subgenomic RNA encoding a C-term NP truncation, and this feature specifically enhances viral fitness during experiments with separation of function recombinant viruses. A second conclusion is that this NP variant binds dsRNA and inhibits dsRNA immune responses, dependent upon the ability to bind dsRNA. These observations are well supported and are worthy of publication.

The third conclusion is that N*M210 inhibits stress granule formation and that this activity is the basis of improved viral fitness. While this may be true, one would need additional experiments to be convinced of this mechanistic interpretation, as detailed below. This conclusion should be modified.

This review is from Roy Parker and I would be willing to clarify these comments for the authors if needed.

Specific comments:

1) The biggest issue with this manuscript is the interpretation that the fitness advantage of the M120 protein comes, at least in part, from inhibiting stress granules. As detailed below, I have several issues with this conclusion. My suggestion is that authors note the impacts of Nfl and NM120 on stress granules and P-bodies but not make any mechanistic conclusions for the reasons detailed below.

A) The critical experiment arguing that stress granules affect viral fitness in a manner influenced by NM120 is that the fitness difference in TRS+ (which makes NM120) and TRS- (which does not) is that the difference between these two viruses is less in a g3bp1∆ cell line when compared to WT cells (Figure 6J and K). However, I am concerned about this experiment since:

i) The difference in fitness between recombinant viruses in g3bp1∆ cell line is observed only at 72hpi, which raises the concern that the result is not robustly different. What is the model for N*210 playing a role only at late infection time points?

ii) A g3bp1∆ cell line will typically express G3BP2 and therefore still make stress granules, which would limit the conclusion that this effect is due to stress granules per se.

iii) In principle, the differences between a WT and g3bp1∆ cell line can be due to differences in G3BP1 function as opposed to stress granules per se.

iv) We often observe differences in CRISPR edited cell lines that are not due to the targeted deletion and therefore using a resuce construct is important to conclude that the differences are due to the g3bp1∆ per se.

B) I do think it is not great to consider any G3BP1 puncta as stress granules when it is now clear that in cells activating RNase L the assemblies are fundamentally different. If the authors want to continue to mix stress granules and RLBs, they should simply collectively refer to them as G3BP1 puncta.

2) It is notable that the authors observe that Nfl and NM120 inhibits stress granule formation with arsenite in a manner dependent on the dsRNA binding regions and by implication independent of the G3BP interacting domain. It would be important to note this is different from what has been previously described and provide a possible explanation.

3) On many figures (especially quantifications), it would be helpful to have a title with distinguishing experiment characteristics (cell line, specific KO, etc. e.i. 6J and 6K, distinguishing WT and KO cell experiments). The figure panels should also be labeled to make it easy to interpret the figures.

4) Quantifications of blots in figures 1, 2 to measure proportions of N isoforms would be helpful.

5) Many of the IF images have Dapi signal scattered throughout the cytoplasm (see 6A for example). Is this a consequence of transient transfection? I worry about contamination.

Reviewer #2:

The manuscript by Mulloy et al. is built on a previous observation, also reported by others, that certain SARS-CoV-2 variants —including Alpha, Gamma, and Omicron— have acquired two mutations within the N coding sequence (R203K/G204R), resulting in a novel canonical core sequence (CS) (ACGAAC). At the RNA level, this novel CS leads to the synthesis of a new subgenomic RNA (sgRNA-N*), which encides a truncated ORF named N*M210. At the protein level, prior studies have shown that these amino acid substitutions may affect the function of the N protein during infection, particularly itsRNA binding activity.

A recent study by Mears et al. (PLoS Biol. 2025) demostrated that the truncated N*M210 contributes to viral fitness during infection by antagonizing type I interferon production. This paper also described distinct phenotypes when the N coding sequence was mutated, compared to silent mutations that only ablate the TRS.

Mulloy et al. confirm and extend the findings of Mears et al. by investigating the mechanisms underlying the enhanced fitness of viruses expressing higher levels of the truncated protein N*M210. They hypothesised that N*M210 might sequester dsRNA in the cytoplasm, thereby preventing recognition by host pattern recognition receptors involved in IFN responses. This hypothesis is explored primarily through overexpression experiments assessing the impact on cellular pathways induced by dsRNA.

Additionally, the authors engineered recombinant SARS-CoV-2 viruses either including (KR+TRS) or lacking (KR-TRS) the internal TRS within the N gene to study the relevance of N*M210 in IFNß production and PKR activation during infection. Interestingly, they confirmed in vivo that viruses expressing N*M210 exhibit superior fitness during infection of K18-hACE2 transgenic mice.

The manuscript provides novel insights into the mechanisms driving the fitness advantage of KR+TRS viruses that overproduce N*M210. First, they extensively demostrate by in vitro experiments that N*M210 has a greater capacity to bind dsRNA than the full-length N protein. Furthermore, the superior fitness of KR+TRS viruses persists even in the absence of IFN responses, suggesting additional mechanisms beyond IFNß suppression. Then, they explore the formation of SGs, which have been associated with antiviral responses, using overexpression experiments.

While the study advances our understanding of the mechanisms of enhanced virus fitness when N*M210 is expressed during infection, many conclusions are based on overexpression experiments. To strengthen the physiological relevance, further validation using viral infection models is necessary.

The discussion should clearly distinguish between results obtained from overexpression versus infection experiments, as this distinction is critical for interpreting the findings and assess their physiological relevance.

Specific points

1. Line 331. "Despite the superior ability of N*M210 to bind and co-localize with dsRNA relative to NFL, in these assays we did not observe a significant difference in the ability of these two constructs to block dsRNA-induced responses". It is intriguing that overexpression assays (Fig. 5) show no significant differences in innate immune responses depending on dsRNA binding, such as phosphorylation of PKR and IRF3 and activation of RNaseL, between N*M210 and NFL, despite their differing dsRNA-binding capacities. Could the authors elaborate on this discrepancy?

2. During authentic viral infection, increase din IFNß production was observed when N*M210 was expressed (Fig. 5N), consistents with previous findings. However, no significant differences were observed in PKR phosphorylation (Fig. 5M), suggesting that downstream processes dependent on PKR phsphorylation, such as SG formation, may also be unaffected. The proposed mechanism of inhibition of SG formation should be confirmed in infected cells.

3. Line 336. The statement "Both viruses induced an upregulation in total PKR and pPKR levels; however, pPKR levels were higher in cells infected with KR-TRS (Fig 5L & M)" should be revised. According to Fig. 5M, infection with the KR+TRS virus does not increase PKR phosphorylation compared to mock-infected cells, whereas a significant increase was observed when the TRS was ablated (KR-TRS).

4. Some of the antiviral responses induced by dsRNA, such as PKR and RNaseL activation, lead to translation inhibition. It would be valuable to assess whether translation is differentially affected during KR+TRS vs KR-TRS infection.

5. Line 384. Based on overexpression experiments, the authors conclude that "Given that N*M210 blocks dsRNA-induced SGs (Fig 6) and inhibits PKR activation (Fig 5), and both phenomena required the dsRBM, we predicted that N*M210 achieves both via sequestering the SG inducer, dsRNA, preventing PKR activation and subsequent SG formation". However, both N*M210 and N-FL inhibit PKR phospholylation to a similar extent (Fig. 5C), while only N*M210 inhibits polyI:C-induced SG formation. This point warrants further discussion.

6. While overexpression experiments show that N*M210 inhibits polyI:C-induced SG formation, its relevance during infection remains unconfirmed. Co-infection competition assays in G3BP1 KO cells, which are impaired for SG formation, suggest forshowed a decrease in the fitness advantage of KR+TRS virus, suggesting that SG inhibition may contribute to viral fitness. However, additional infection-based evidence is needed.

Minor points

1. Line 84. "sgRNA synthesis is governed by the presence of short ~6 nucleotide sequences, called transcription regulatory sequences (TRSs)". This sentence and the corresponding figure (Fig. 1) should be revised. The 6-nt conserved sequence is the core sequence (CS), while TRSs include the CS and 5' and 3' flanking sequences, which also regulate transcription.

2. While many viruses inhibit SGs and P-bodies during infection, some transiently induce SGs or promote their formation to enhance replication, such as RSV (Linquist et al., 10.1128/JVI.00260-10). This should be acknowledged.

3. Line 564. The discussion highlights the reverse genetics approach used to dissect transcriptional versus protein-level effects. However, Mears et al. (PLoS Biology 2025) previously addressed this using a similar strategy and should be cited.

---

## [Decision Letter · Decision Letter 2]

7 Jan 2026

Dear Jenn,

Thank you for your patience while we considered your revised manuscript "A truncated SARS-CoV-2 nucleocapsid protein enhances virus fitness by evading antiviral responses" for publication as a Research Article at PLOS Biology. This revised version of your manuscript has been evaluated by the PLOS Biology editors, the Academic Editor and the original reviewers.

Based on the reviews, we are likely to accept this manuscript for publication, provided you satisfactorily address the remaining points raised by the reviewers. Please also make sure to address the following data and other policy-related requests.

1) We routinely suggest changes to titles to ensure maximum accessibility for a broad, non-specialist readership, and to ensure they reflect the contents of the paper. In this case, we would suggest a minor edit to the title, as follows. Please ensure you change both the manuscript file and the online submission system, as they need to match for final acceptance:

“Evolution of a truncated nucleocapsid protein enables SARS-CoV-2 to suppress dsRNA-triggered antiviral defenses”

2) Please add weblink to the funding agencies in the Financial Disclosure statement in the manuscript details.

3) The Ethics statement needs to be a separate, independent (and the first) subheading in the Material & Methods section. It must include the full name of the IACUC/ethics committee that reviewed and approved the animal care and use, as well as the protocol/permit/project license number. https://journals.plos.org/plosbiology/s/ethical-publishing-practice

Please supply the numerical values either in the a supplementary file or as a permanent DOI’d deposition for the following figures:

Figure 1GI, 2DF-M, 3BDFG, 4E, 5BCDFGHMNO, 6BEFGJ, 7CE-J, 8BCD, S1A-D, S2BCE-M, S3C, S4B, S5BCD, S6AB

4) Please cite the location of the data clearly in all relevant main and supplementary Figure legends, e.g. “The data underlying this Figure can be found in S1 Data” or “The data underlying this Figure can be found in https://doi.org/10.5281/zenodo.XXXXX”

5) We require the original, uncropped and minimally adjusted images supporting all blot and gel results reported in the Figures 1DEFHJK, 2BC, 3A, 4AF, 5AEIJL, 6H, S1EFG, S2AD, S5E

6) We will require these files before a manuscript can be accepted so please prepare and upload them now. Please carefully read our guidelines for how to prepare and upload this data: https://journals.plos.org/plosbiology/s/figures#loc-blot-and-gel-reporting-requirements

7) Please add a scale bar in all microscopy pictures

8) Please ensure that your Data Statement in the submission system accurately describes where your data can be found and is in final format, as it will be published as written there

We expect to receive your revised manuscript within two weeks.

*Published Peer Review History*

*Press*

Sincerely,

Melissa

Melissa Vazquez Hernandez, Ph.D.

Associate Editor

PLOS Biology

REVIEWERS' COMMENTS

Reviewer #1:

This revised manuscript is improved and the authors have generally addressed our prior comments. I am supportive of publication. However, I recommend the authors improve the manuscript in two manners.

1) I recommend a paragraph that clarifies and functions of N*M120 and how it might function. For example:

The N*M120 protein could improve viral fitness by at least three different mechanisms. First, because N*M120 inhibits the activation of RNAse L and PKR, which are both antiviral, the inhibition of these antiviral responses could promote viral replication. Second, because N*M120 also can inhibit stress granule formation triggered by sodium arsenite, N*M120 could promote viral fitness by blocking RNP granule formation. Finally, N

*M120 could promote viral fitness by limiting the interferon response, although because VERO cells, which are deficient in the interferon response, still show some response to N*M120 this is unlikely to be the only antiviral activity of N*M120.

2) I recommend that the authors acknowledge the limitation of the lack of G3BP1 rescue in the g3bp∆∆ cells more clearly. I recommend this addition because we observe that these same g3bp∆∆ cells show enhanced replication of OC43, but this is not complemented by a WT G3BP1 transgene (unpublished observation). Thus, without a rescue experiment, I would be cautious about the interpretation of the g3bp∆∆ cells.

Reviewer #2:

The authors have satisfactorily addressed most of this reviewer's comments.

They have included additional experimental results and further discussion, which strengthen the relevance of the manuscript.

There are a few minor points that should be addressed.

Specific points

1. Line 446. The authors conclude that "KR+TRS was more effective at inhibiting G3BP1 and TIAR foci formation than KR-TRS". These results, shown in Fig 7C and 7D, require a statistical analysis to confirm their significance.

2. Line 634. The authors suggest that "N*M210 should localize to the nucleus. In agreement with this, we observe that a fraction of N*M210 adopts nuclear localization after ectopic expression (Fig 3C)". It should be clarified that the presence of N*M210 has not been confirmed in the context of infection. Its detection in the nucleus after ectopic expression might be the a consequence of overexpression. Indeed, according to Fig. 3A, a higher amount of N*M210 is detected in transfected cells compared to NFL.

---

## [Editor Report · Decision Letter 3]

26 Jan 2026

Dear Jen,

Thank you for the submission of your revised Research Article "Evolution of a truncated nucleocapsid protein enhances SARS-CoV-2 fitness by suppressing antiviral responses" for publication in PLOS Biology. On behalf of my colleagues and the Academic Editor, Jason T. Ladner, I am pleased to say that we can in principle accept your manuscript for publication, provided you address any remaining formatting and reporting issues. These will be detailed in an email you should receive within 2-3 business days from our colleagues in the journal operations team; no action is required from you until then. Please note that we will not be able to formally accept your manuscript and schedule it for publication until you have completed any requested changes.

PRESS

Sincerely,

Melissa

Melissa Vazquez Hernandez, Ph.D., Ph.D.

Associate Editor

PLOS Biology
